# Consistent signatures in the human gut microbiome of old- and young-onset colorectal cancer

Youwen Qin [1,2,7] ✉, Xin Tong [1,7], Wei-Jian Mei [3,7], Yanshuang Cheng[3], Yuanqiang Zou [1,4], Kai Han [3], Jiehai Yu [3], Zhuye Jie[1], Tao Zhang [1,5,6], Shida Zhu[2], Xin Jin [1], Jian Wang[1], Huanming Yang [1], Xun Xu [1], Huanzi Zhong [1,2], Liang Xiao [1,4] & Pei-Rong Ding [3] ✉

The incidence of young-onset colorectal cancer (yCRC) has been increasing in recent decades, but little is known about the gut microbiome of these patients. Most studies have focused on old-onset CRC (oCRC), and it remains unclear whether CRC signatures derived from old patients are valid in young patients. To address this, we assembled the largest yCRC gut metagenomes to date from two independent cohorts and found that the CRC microbiome had limited association with age across adulthood. Differential analysis revealed that well-known CRC-associated taxa, such as *Clostridium symbiosum*, *Peptostreptococcus stomatis*, *Parvimonas micra* and *Hungatella hathewayi* were significantly enriched (false discovery rate <0.05) in both old- and young-onset patients. Similar strain-level patterns of *Fusobacterium nucleatum*, *Bacteroides fragilis* and *Escherichia coli* were observed for oCRC and yCRC. Almost all oCRC-associated metagenomic pathways had directionally concordant changes in young patients. Importantly, CRC-associated virulence factors (*fadA*, *bft*) were enriched in both oCRC and yCRC compared to their respective controls. Moreover, the microbiome-based classification model had similar predication accuracy for CRC status in old- and young-onset patients, underscoring the consistency of microbial signatures across different age groups.

Colorectal cancer (CRC) is one of the most common non-sex-specific cancer worldwide, after lung cancer, and accounts for about one million deaths in 2020[1]. In the last few decades, the incidence of CRC has remained stable or decreased in the developed countries[2]. However, the number of young-onset CRC (yCRC, colorectal cancer diagnosed in patients under the age of 50 years) has been increasing globally[3]. The striking number of young-onset patients is a growing challenge in CRC management and global health. Despite up to 34% of yCRC having a family history of colorectal cancer[3], the majority of cases are without clear genetic factors. The genetic background of world population is unlikely changed over the last several decades. The increasing incidence of yCRC may be attributed to changing environmental and lifestyle factors.

Among environmental factors, the gut microbiome—the microbial ecosystem residing primarily in the large intestine—has been implicated in colorectal carcinogenesis. Animal studies have pinpointed

[1]BGI Research, Shenzhen 518083, China. [2]BGI Genomics, Shenzhen 518083, China. [3]Department of Colorectal Surgery, Sun Yat-sen University Cancer Center; State Key Laboratory of Oncology in South China; Guangdong Provincial Clinical Research Center for Cancer, Guangzhou 510060, China. [4]Shenzhen Engineering Laboratory of Detection and Intervention of Human Intestinal Microbiome, Shenzhen, China. [5]Shenzhen Key Laboratory of Human commensal microorganisms and Health Research, Shenzhen, China. [6]BGI Research, Wuhan 430074, China. [7]These authors contributed equally: Youwen Qin, Xin Tong, Wei-Jian Mei. ✉e-mail: qinyouwen@genomics.cn; dingpr@sysucc.org.cn

**Fig. 1 | Limited association between the gut microbiome and age in CRC patients. a** The number of patients in the Guangzhou cohort stratified by age and sex. **b** Two-dimension scatter plot shows the overall pattern of samples. Principle coordinate analysis (PCoA) was performed based on the Bray–Curtis distance calculated from the abundance profile at species level. Each point represents one sample, and color scale indicates age. Samples from female and male patients are in triangles and squares, respectively. Scatterplots of relationship between age and PCoA axis 1 (**c**), PCoA axis 2 (**d**), number of species (**e**), and Shannon index (**f**). The correlation coefficient was calculated using the Spearman method. The solid red line was fitted by smooth function in R, and the gray area is the 95% confidence interval. The Shannon index was calculated based on the abundance profile at the species level.

three prominent examples of gut microbial toxins in colorectal carcinogenesis[4,5]. For example, *Bacteroides fragilis* toxin promotes colon tumorigenesis by activation of the $T_H 17$ cell response[6]. Additionally, *Fusobacterium nucleatum* adhesion protein A (FadA) can bind to E-cadherin of CRC cells, activates beta-catenin signaling, and regulates oncogenic responses[7]. Certain strains of *Escherichia coli* produce colibactin, a small-molecule genotoxin that can adduct to DNA and induce double-strand DNA breaks[8]. In humans, about a dozen metagenomic studies on various populations across the world have identified substantial changes in abundance of specific bacteria[9–16]. Meta-analyses of these datasets have identified globally cross-cohort microbial signatures that can predict CRC at high accuracy[13,14,17]. However, the studied cohorts have predominantly consisted of old-onset patients. The number of yCRC patients in these studies varied from 0 to 28, accumulating to 72 in total (Supplementary Table 1). It is uncertain whether these microbial signatures are specific to oCRC or can be generalized to yCRC.

A very recent study using 16S rRNA gene sequencing reported distinct dysbiosis in the human gut microbiome of yCRC patients[18]. Although they validated their findings in a subset of individuals using metagenomic sequencing, their main discoveries were based on 16S rRNA data. The technical limitations of this approach have limited the ability to draw definitive conclusions[19]. Further efforts based on metagenomic sequencing in larger cohort are needed to explore the gut microbial signatures in yCRC. Deep metagenomic sequencing can be leveraged to investigate strain-level diversity, providing valuable insights for experimental validation.

In this study, we generated 460 CRC stool metagenomes, including data from 167 yCRC patients. By integrating these data with a publicly available yCRC dataset, we identified consistent microbial signatures in both oCRC and yCRC. These signatures encompassed well-known CRC-associated taxa and virulence factors. Our strain-level analysis, focusing on three CRC-associated species (*F. nucleatum*, *B. fragilis* and *E. coli*), supported the concordance in oCRC and yCRC. We further leveraged other publicly available CRC metagenomic datasets and demonstrated that the microbiome-based predictive models had similarly high accuracy in both oCRC and yCRC patients. These results provide valuable insights into the generalizable microbial signatures of CRC and expand our understanding of CRC microbiome.

## Results

We recruited 460 CRC patients from a single hospital in Guangzhou (Methods). All patients were treatment naïve by the time of enrollment. Our cohort included patients with a wide age range, from 21 to 88 years old (Fig. 1a), with 95 patients diagnosed under the age of 40 and 167 patients under the age of 50. Across all age groups (Supplementary Data 1), there were more male patients than female patients. 14.8% (n = 68) of cancers were stage I, 32.0% (n = 147) stage II, 36.1% (n = 166) stage III and 17.2% (n = 79) stage IV; 24.8% (n = 114) were from the right hemicolon, 34.8% (n = 160) left hemicolon and 40.4% (n = 186) rectum; 14.6% (n = 67) were with family history of CRC. There was no correlation between incidence age and sex (P = 0.06), tumor stage (P = 0.10), tumor location (P = 0.13), and family history of CRC (P = 0.3).

We observed a weak correlation between body mass index (BMI) and age (Pearson correlation coefficient = 0.12, $P = 0.01$).

## Limited association between the gut microbiome and age in CRC patients

The relationship between age and the gut microbiome in CRC patients was investigated using shotgun metagenomic sequencing of stool samples. We generated a total of 32,403 million paired-end high-quality reads, with an average of 70 million paired-end reads per sample (Methods). We found no correlation between age and alpha-diversity, defined as the number of observed species and Shannon index (Fig. 1e, f). The adjustment of confounders (BMI, sex, tumor location and stage, and smoking) did not increase the association between alpha-diversity and age. Similarly, there was no correlation between age and the first two coordinates of the principal coordinate analysis (PCoA) (Fig. 1b–d). This was supported by the permutation multivariate analysis of variance (PERMANOVA) test, showing that age only explained a small fraction of microbiome variance ($R^2 = 0.003$, $P = 0.15$).

To identify specific taxa associated with age, we tested the correlation between age and species abundance (Methods). We only found that the abundance of four species, namely *Prevotella stercorea*, *Bifidobacterium dentium*, *Prevotella copri*, and *Prevotella bivia*, were significantly correlated with age (false discovery rate (FDR) adjusted $P < 0.05$, Supplementary Data 2). The associations of *Prevotella* species were negative, while that of *B. dentium* was positive. *B. dentium* and *P. bivia's* associations with age were independent of body mass index (BMI), sex, tumor location and stage, family history of CRC, and smoking. Although *Prevotella* species are commonly found in the human gut microbiota and have been linked to dietary habits, their relative abundances were age-dependent and dropped from adulthood to old age[20]. *B. dentium*, which is commonly found in the human oral microbiome[21], had increased abundance and prevalence in the gut with age[22]. While two dozen of other bacterial species have been identified as age-associated[23], their associations with age in CRC patients were weak, suggesting that CRC status may outperform age in shaping the gut microbiome.

As CRC is one of the most studied traits in gut microbiome research, a collection of CRC-associated taxa has been robustly identified in previous studies. We obtained a list of 118 CRC-associated taxa from gutMDisorder[24] (Methods). Among them, 24/25 taxa reported in at least two studies were detected in our cohort with prevalence rates from 15.43% to 90.65% (average 58.41%), with 16 taxa presented in over half patients. Importantly, only the abundance of *P. copri* was correlated with age, but its correlation was not significant after adjustment for confounders (Supplementary Data 2).

Additionally, we validated our findings in a recently published yCRC cohort (Fudan cohort)[18]. In this cohort, age was given as a binary variable, old or young, with the cutoff of 50. Consistently, we separated our Guangzhou patients into old (age ≥ 50) and young (age < 50) groups. We only found nine species with differential abundance ($P < 0.05$) between old and young groups in both cohorts (Supplementary Data 3). Four of them (*P. stercorea*, *B.dentium*, *P. copri* and *P. bivia*) were mentioned above. Although the other five species included previously reported CRC-associated microbe *Alistipes indistinctus*[10,15], and CRC-depleted microbes *Eubacterium rectale*[10] and *Faecalibacterium prausnitzii*[10], none passed the multiple testing correction (FDR adjusted $P > 0.05$). Taken together, there was limited (if any) association between the known CRC-associated taxa and age.

## Bacterial species associated with oCRC and yCRC

To investigate the gut microbiome changes in oCRC and yCRC patients, we compared them to age-matched controls. We reanalyzed the stool metagenomic data from the Fudan cohort[18] and integrated it with our Guangzhou cohort (Methods). In accordance with Yang et al.[18], the yCRC was defined as age under 50 years old, and the others

were oCRC. A PCoA based on species-level abundance showed that disease effect surpassed the batch effect (Fig. 2a–c). The Guangzhou patients had similar distributions in PCoA1 and PCoA2 with patients in the Fudan cohort, and lower median values than the controls. Furthermore, the CRC status explained a slightly higher variance than study effect ($R^2 = 0.00408$ and 0.00376, PERMANOVA). Therefore, the batch effect in Guangzhou and Fudan cohorts was limited.

Previous studies have suggested that the microbiome of CRC patients had a higher alpha diversity than controls, possibly due to the expansion of typically oral microbes in addition to the baseline gut microbiome[13,14,17]. We confirmed this finding in the Fudan cohort, where oCRC and yCRC patients had a higher Shannon index than their controls (Fig. 2d). Guangzhou patients also had a higher Shannon index than the Fudan controls, supporting the increased diversity in CRC patients.

To identify taxa that are differentially abundant in CRC patients, we conducted two sets testing. The first set of testing was conducted only on the Fudan cohort, while the second set was conducted on the Guangzhou patients and Fudan controls (Methods). Our analysis revealed four species (*Clostridium symbiosum*, *Peptostreptococcus stomatis*, *Parvimonas micra,* and *Hungatella hathewayi*) that were consistently enriched (FDR adjusted $P < 0.05$) in oCRC and yCRC patients in both cohorts compared to Fudan controls (Fig. 2e, Supplementary Data 4). These four species are well-known CRC-associated biomarkers[13,14] and were not associated with age (Supplementary Data 3). *C. symbiosum*, for instance, was first reported by a qPCR study[25] and confirmed in a meta-analysis study that integrated five shotgun metagenomic studies[13]. *P. micra* and *P. stomatis* were among the most important features in CRC classifiers built on stool microbiome data[13].

We also found six other microbial species that were differentially abundant (FDR adjusted $P < 0.05$) in oCRC and showed similar trends in yCRC in both cohorts (Supplementary Fig. 1, Supplementary Data 4). Among these six taxa, *Eggerthella lenta*, *Erysipelatoclostridium ramosum*, and *Flavonifractor plautii* were enriched in CRC groups and have been previously reported as biomarkers for CRC[10,18]. In particular, *F. plautii* was identified as a biomarker for yCRC by Yang et al.[18]. Two known bacteria *Eubacterium rectale* and *Ruminococcus bicirculans*, as well as a metagenomically assembled taxon *Eubacterium sp. CAG38* was depleted in CRC microbiome. *E. rectale* is one of the most prevalent human gut bacteria[26] and was reproducibly reported with decreased abundance in CRC patients compared to healthy controls[9,10].

On the other hand, three of the four taxa (*Alistipes indistinctus*, *Clostridium aldenense*, *Eisenbergiella tayi,* and *Fusobacterium* sp. oral taxon 370) enriched in yCRC showed similar trends in oCRC as well (Supplementary Fig. 2, Supplementary Data 4). *Fusobacterium* sp. oral taxon 370 is one of the typical oral bacteria linked to CRC in old-onset patients[14]. While *A. indistinctus* was at a low abundance in the human gut microbiome, it was involved in CRC carcinogenesis and treatment response[27]. We specifically analyzed two taxa, *B. fragilis*[6] and *F. nucleatum*[7], with proposed carcinogenesis mechanism. Although the *B. fragilis* abundance was higher in CRC than control, the significance was lost after multiple hypothesis adjustment (Supplementary Fig. 3). *F. nucleatum's* prevalence was surprisingly low in the Fudan cohort, while its abundance had a similar distribution in oCRC and yCRC of the Guangzhou cohort. Additionally, at the significant level of nominal $P < 0.05$, 23 of 24 species passing the threshold had directionally consistent changes in old- and young-onset patients compared with their controls (Supplementary Data 4). Overall, our findings indicate that most of the CRC-associated taxa showed concordant changes in both oCRC and yCRC microbiomes compared to their controls.

## Strain-level diversity of *F. nucleatum*, *B. fragilis,* and *E. coli*

Our deep metagenomics sequencing data allowed us to investigate the strain-level insights into CRC-associated species. Given the inherent

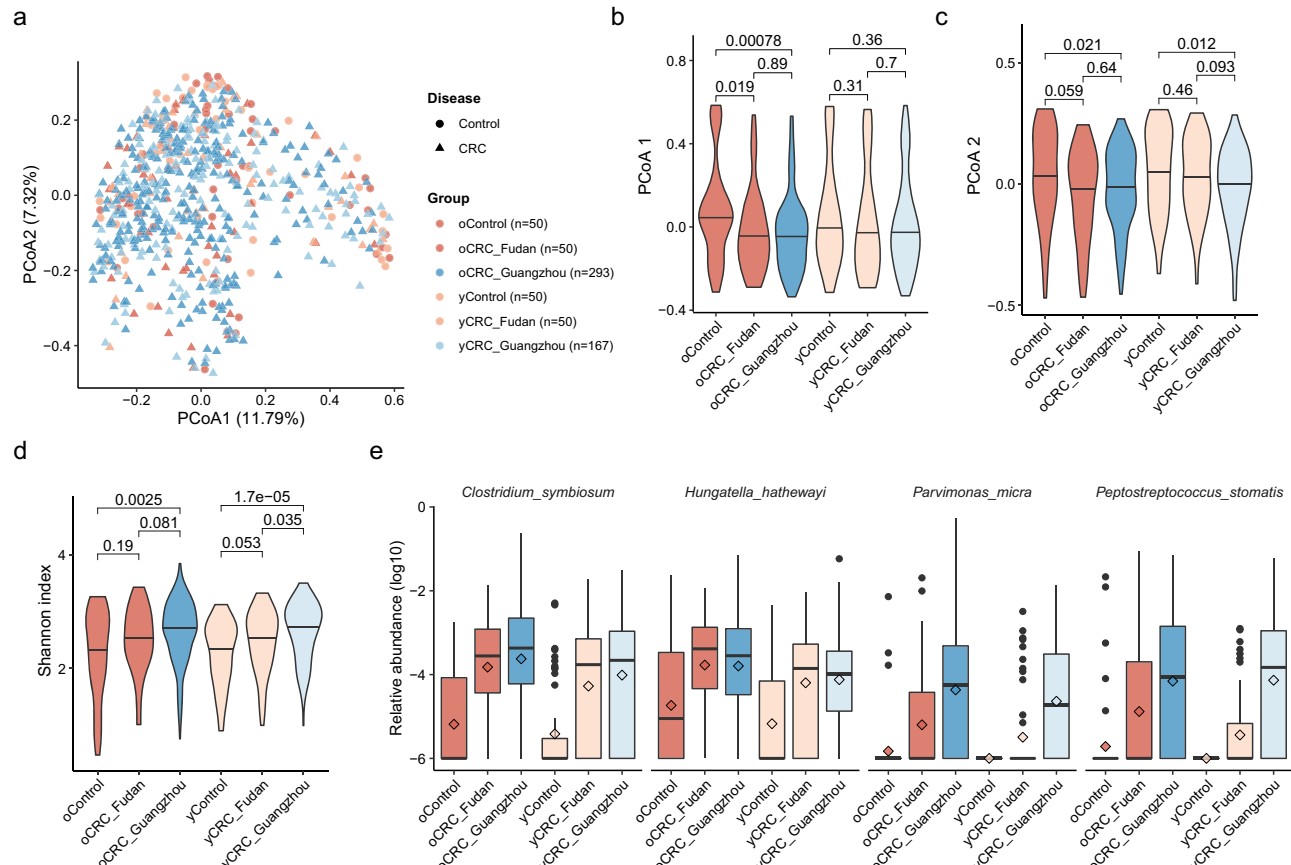

**Fig. 2 | Consistent changes of CRC-associated microbes in old- and young-onset patients in two independent cohorts. a** Two-dimension scatter plot shows the overall distribution of Fudan and Guangzhou samples. Principle coordinate analysis (PCoA) was performed based on the Bray–Curtis distance calculated from the abundance profile at species level. Each point represents one sample. Samples from Fudan and Guangzhou cohorts are in red and blue, respectively. Circles are control samples, while triangles are CRC samples. Violin plots show values of PCoA axis 1 (**b**) PCoA axis 2 (**c**), and Shannon index (**d**) across different groups. The thick horizon line indicates the 50% percentile. *P* values on the top were calculated by two-side Wilcoxon rank-sum test. **e** Four well-known CRC-enriched species significantly enriched in both oCRC and yCRC patients at false discovery rate (FDR) adjusted *P* < 0.05. The sample size of each group is the same as **a**. The relative abundance is in log$_{10}$ scale and zeros were replaced by a small value. The box plots show the median (thick line), interquartile range (box limits), 1.5× the interquartile range span (whiskers), and outliers (dots). Diamond shape indicates the mean abundance.

challenges of strain level, we focused on three CRC-associated species *F. nucleatum (Fn), B. fragilis (Bf),* and *E. coli (Ec)* in the Guangzhou cohort. We used StrainPhlAn3[17] to construct the phylogenetic tree and used inStrain[28] to examine the genome-wide sequence diversity (Methods).

*Fn* was identified in 63 samples according to its marker genes and the corresponding phylogenetic tree showed no correlation with patient age, tumor stage, and location (Supplementary Fig. 4a). No significant difference was observed in the *Fn* prevalence between oCRC and yCRC. Genome-wide sequence analysis revealed similarly high population-level average nucleotide identity (popANI)[28] values to the *Fn* reference genome in oCRC and yCRC metagenomes, with no significant difference (Supplementary Fig. 4b). We also evaluated the nucleotide diversity, an indicator of strain diversification. Our analysis revealed no association between *Fn* nucleotide diversity and patient age, tumor stage or location (Supplementary Fig. 4c–e). A study reported that *F. animalis (Fa,* also known as *Fn* subspecies *animalis)* had higher abundance and prevalence than *Fn* in tumor samples[29]. In line with this finding, we found *Fa* in more samples with higher coverage than *Fn* in our cohort (Fig. 3). *Fa* prevalence, popANI, and nucleotide diversity values showed no difference in oCRC and yCRC (Supplementary Fig. 5). Taken together, the analysis of *Fn* and *Fa* distribution and diversity revealed no distinctions between oCRC and yCRC. However, we noted a higher nucleotide diversity of *Fa* in the

colon compared to rectum tumors (Supplementary Fig. 5). A similar trend was observed for *Fn*, albeit with lower significance due to a smaller sample size. This observation suggests that *Fn* and *Fa* exhibited increased diversity in patients with colon tumors.

For *Bf*, we identified two distinct phylogenetic clusters (Supplementary Fig. 6a). Cluster 1 (*N* = 267) was dominated by samples with metagenome-assembled genomes (MAGs) annotated to strain NCTC9343 (average nucleotide identity (ANI) > 95%), while cluster 2 (*N* = 67) was dominated by samples with MAGs annotated to strain Q1F2 (Methods). The distribution of oCRC and yCRC was not different in the two clusters. The phylogenetic tree revealed no correlation with patient age, tumor stage or location, within or between clusters (Supplementary Fig. 6a). Genome-level analysis demonstrated high popANI values to the reference genomes in oCRC and yCRC (Supplementary Fig. 6b). While the nucleotide diversity of cluster 1 was not associated with age, the nucleotide diversity of cluster 2 was higher in yCRC than oCRC (*P* = 0.02, Supplementary Fig. 6c). This suggests that *Bf* strain in cluster 2 samples may be under stronger selection pressure in yCRC patients.

For *E. coli*, our analysis identified only one strain cluster (ANI > 95% to strain ATCC 11775) in 317 (69%) samples, with no significant difference in prevalence between oCRC and yCRC. The marker gene-based phylogenetic tree analysis revealed no correlation with patient age, tumor stage or location (Supplementary Fig. 7a). Genome-level

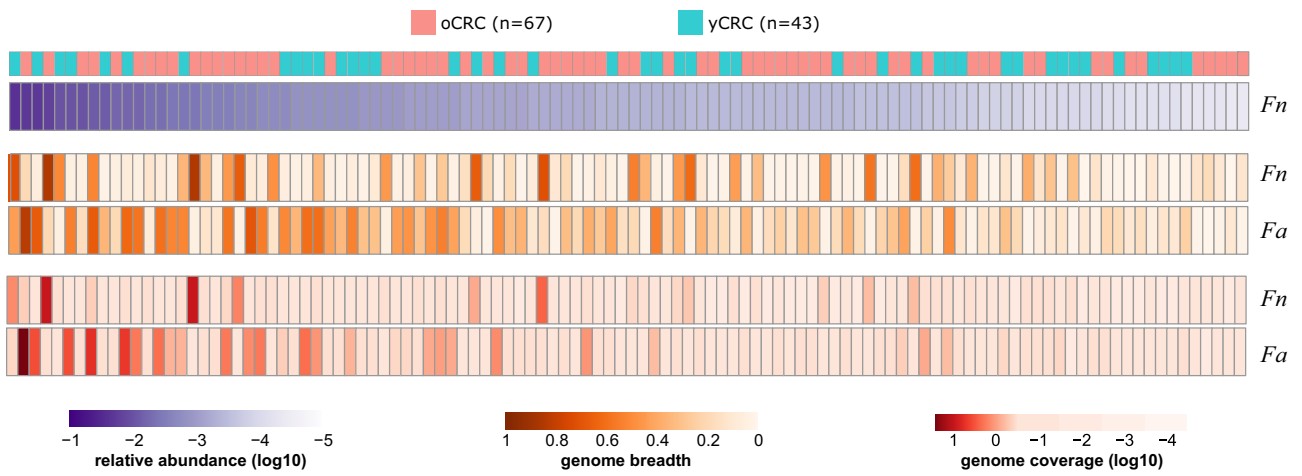

**Fig. 3 | Higher prevalence and abundance of *F. animalis* than *F. nucleatum* in CRC patients.** Heatmap shows the abundance, genome-wide breadth, and coverage of *F. nucleatum* (*Fn*) and *F. animalis* (*Fa*). Only 110 samples which had genome breadth >0.1 and coverage >0.1 for *Fn* (RefSeq GCF_008633215.1) or *Fa* (RefSeq GCF_000158275.2) reference genomes were included. Samples were sorted in decreasing order by the relative abundance of *Fn*, calculated by MetaPhlAn 3.

analysis indicated a similar popANI value to the reference genome in oCRC and yCRC (Supplementary Fig. 7b). Nucleotide diversity analysis showed no association with patient age, tumor stage or location (Supplementary Fig. 7c–e).

## Functional metagenomic signatures for oCRC and yCRC

Unlike 16S rRNA gene amplicon data, metagenomes allow us to access the functional capacity of the gut microbiome. In our Guangzhou cohort, age did not have a significant association with the top two axis of PCoA, calculated from the microbial pathway profile (Supplementary Fig. 8). In PERMANOVA, age explained a small and non-significant amount of overall variance of the microbial pathway variation ($R^2 = 0.005$, $P = 0.13$). To identify specific microbial pathway associated with age, we tested the associations between age and abundances of metaCyc[30] pathways (Methods). Only one metaCyc pathway, PWY-6608: guanosine nucleotides degradation III, was associated with age at $P < 0.01$ with and without adjustment for confounding factors (Supplementary Data 5).

To identify functional signatures associated with oCRC and yCRC, we integrated the functional pathway profiles of our Guangzhou cohort and the published Fudan cohort. PCoA showed that Guangzhou patients were closer to Fudan patients than controls (Supplementary Fig. 9). In addition, CRC status explained higher variance than the study effect ($R^2 = 0.014$ vs. 0.006, PERMANOVA). We then conducted differential analysis in metaCyc pathways (Methods). Out of the 435 tested metaCyc pathways, 69 had differential abundance between oCRC and age-matched controls in both cohorts (FDR adjusted $P < 0.05$, Supplementary Data 6). Among these, while only one pathway, PWY-7316: dTDP-N-acetylviosamine biosynthesis, was differential in yCRC and their controls at the same significant level, the majority (60/69) had concordant enrichment direction in yCRC.

We next examined the well-known CRC-associated microbial *cutC* gene, which encodes choline trimethylamine-lyase responsible for production of the disease-associated trimethylamine (TMA). In total, 113 UniRef90[31] gene families annotated as *cutC* orthologs were detected in at least one sample in Guangzhou or Fudan cohorts. Of these, 8 *cutC* orthologs presented in more than half of Guangzhou patients had concordant enriched direction in oCRC and yCRC compared to their controls (Supplementary Data 7). Remarkably, two *cutC* orthologs with represented sequences from *Bacteroides* species, including *Bf* ), reached a statistically significant level of nominal $P < 0.05$. The sum abundance of *cutC* gene families was higher in oCRC and yCRC compared to their controls (Supplementary Fig. 10).

We accessed CRC-associated virulence factors and toxins, focusing on *fadA* (encodes *Fn* adhesion protein A)[7], *bft* (encodes *Bf* enterotoxin)[5], the *pks* genomic island (encodes colibactin in some *Ec* strains)[8], and the *bai* operon (encodes enzymes for the conversion of primary to secondary bile acids in *Clostridium* species)[32] (Methods). *fadA* exhibited significant enrichment in both oCRC and yCRC compared to their respective controls (Fig. 4a). In the strain-level analysis, we identified *Fn* and *Fa* in our samples. Here we further explored *fadA* abundance in the context of *Fn* and *Fa*. *fadA* abundance was not different in samples with either *Fn* or *Fa* (Fig. 4b). Samples with both *Fn* and *Fa* had the highest *fadA* average abundance, while samples without any strain exhibited the lowest *fadA* average abundance. For *bft*, we observed an enrichment trend in CRC compared to controls (Fig. 4a). Notably, both *bft* abundance and prevalence were differentiated in the two phylogenetic clusters defined in the strain-level analysis section (Fig. 4c). The *pks* abundance exhibited variability between cohorts, being higher in CRC than controls in the Fudan cohort but lower in the Guangzhou CRC than controls (Fig. 4a). Strain-level analysis identified strain *Ec* NCTC9343 was the dominant strain in the Guangzhou cohort, and the reference genome of this strain did not contain the *pks* island. We explored the correlation between *Ec* and *pks* abundances, finding a positive correlation in the Guangzhou cohort but no correlation in the Fudan cohort (Fig. 4d). This discrepancy suggests a potential difference in the ecological context between populations. For *bai*, its abundance was significantly higher in yCRC than in the respective controls (Fig. 4a). In the Fudan cohort, oCRC and their controls had similar levels of *bai*, which were higher than those in young controls. This finding aligns with reports of elevated bile acid metabolism in elderly people[33], indicating that aging may influence the association between *bai* and CRC in the elderly. In summary, CRC-associated virulence factors (*fadA*, *bft*) were enriched in both oCRC and yCRC.

## Associations between microbial markers and CRC characteristics

CRC is heterogenous and has diverse molecular characteristics. Here, we investigated the associations between 18 taxonomic markers (those highlighted in Fig. 2, Supplementary Figs. 1–3) and CRC molecular characteristics, including tumor stage and location, mismatch repair (MMR), BRAF and HER2 mutation status (Methods, Supplementary Data 8). Higher abundance of *P. stomatis* was observed in stage III patients, while *E. rectale* showed increased abundance in stage II patients, both compared to stage I patients. Additionally, the

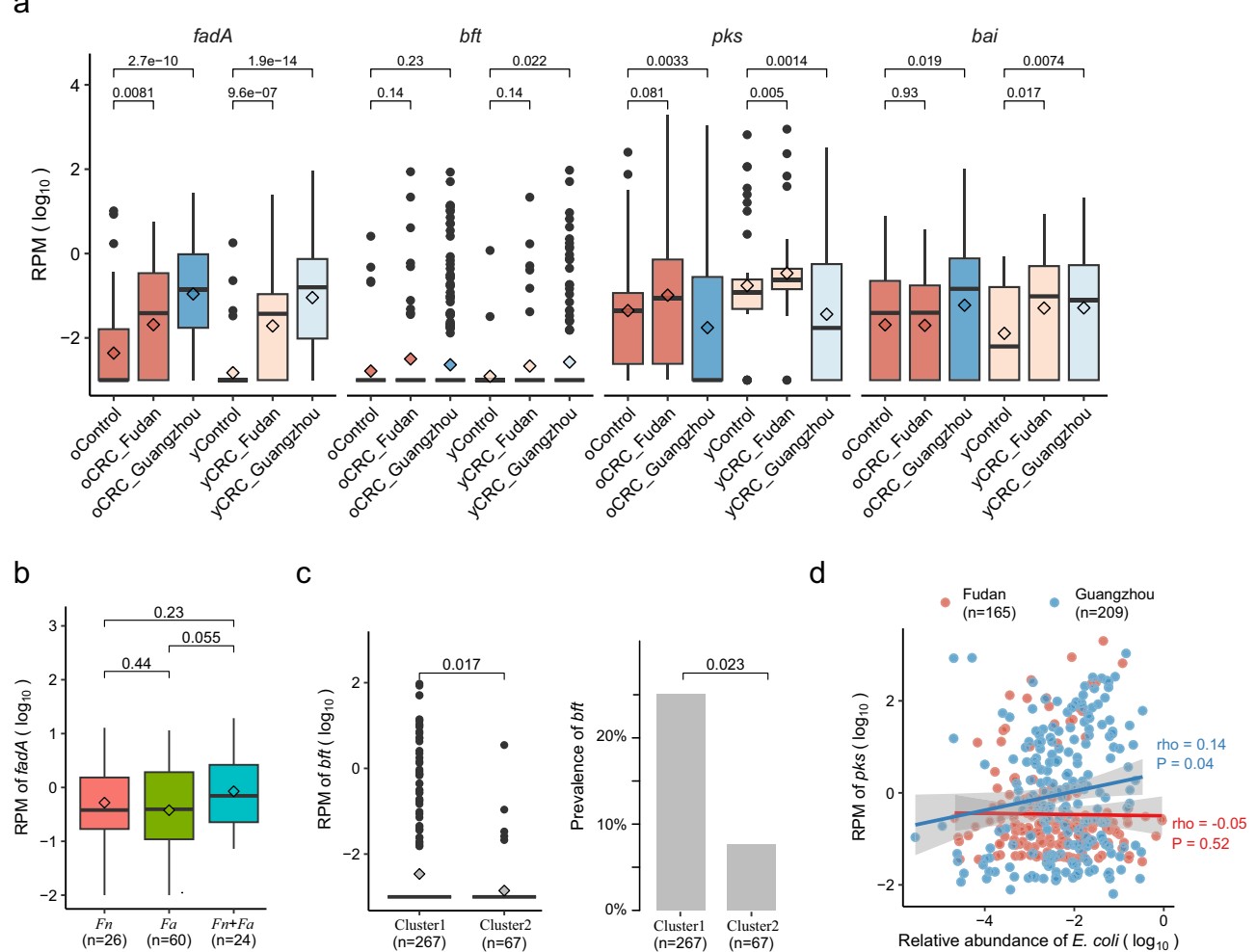

**Fig. 4 | Enrichment of CRC-associated virulence factors and toxins in old- and young-onset patients in two independent cohorts. a** Normalized log abundance of CRC-associated virulence factors in different groups. RPM means reads per million mapped reads, see Methods for gene quantification. *fadA* encodes *F. nucleatum* adhesion protein A; *bft* encodes *B. fragilis* enterotoxin; the *pks* genomic island encodes enzymes to produce genotoxic colibactin (in *E. coli*); the *bai* operon encodes bile acid-converting enzymes (present in some Clostridiales species). Sample sizes for the compared groups: oControl ($n$ = 50), oCRC_Fudan ($n$ = 50), oCRC_Guangzhou ($n$ = 293), yControl ($n$ = 50), yCRC_Fudan ($n$ = 50), yCRC_-Guangzhou ($n$ = 167). **b** Normalized log abundances of *fadA* stratified by the presence of *F. nucleatum* (*Fn*) and *F. animilis* (*Fa*). Presence was defined as genome breadth >0.1 and coverage >0.1. **c** Normalized log abundance and prevalence of *bft* stratified by *B. fragilis* strain clusters. Cluster assignment was conducted based on marker genes and genome-wide sequence analysis (Methods). *P* values on the top were calculated by two-side Wilcoxon rank-sum test. **d** Correlation between normalized log abundances of *pks* and *E. coli*. The correlation coefficient was calculated using the Spearman method. The solid (blue and red) lines were fitted by smooth function in R, and the gray area is the 95% confidence interval. The boxplot conventions are consistent with the description in Fig. 2.

abundance of *E. ramosum* was lower in patients with rectal tumors, *E. rectale* was higher in patients with left-side tumors, *P. micra* was higher in patients with rectal tumors, relative to their counterparts with right-sided tumors. However, in our trend analysis, we did not observe any statistically significant monotonic relationships of these 18 taxa concerning tumor stage and location.

Notably, the relative abundance of *Fn* (maker-gene-based quantification) was higher in MMR deficient (dMMR) patients than MMR proficient (pMMR) patients, and higher in patients with HER2 overexpression compared to those without overexpression (Supplementary Data 8). We further investigated such associations using genome-based quantification, which can offer strain-level resolution as described in the preceding section. Intriguingly, both *Fn* and *Fa* demonstrated elevated abundance in patients with dMMR and HER2 overexpression, relative to their counterparts (Fig. 5).

We additionally examined the associations between gene markers (*fadA, bft, pks,* and *bai*) and tumor stage, location, as well as MMR, BRAF, and HER2 mutation status. No significant association was identified in our cohort.

## Similar prediction accuracy of CRC status in old- and young-onset patients

Several studies have demonstrated the potential of tailoring the gut microbiome for predicting CRC status[13,14,17]. To access the transferability of classifiers across different patient groups, we employed the random forest algorithm to train machine learning models separately on oCRC and yCRC patients (Methods). Our results revealed promising cross-application performance. The model trained on Fudan oCRC patients exhibited robust predictive capability for Fudan yCRC patients, achieving an area under receiver operator curve (AUROC) of 0.7688, only slightly lower than the cross-validated AUROC of 0.8127 (Fig. 6a). Similarly, the model trained on Fudan yCRC patients performed similarly to the cross-validation on oCRC patients, with an AUROC of 0.7548 and 0.7671, respectively (Fig. 6a). We extended our

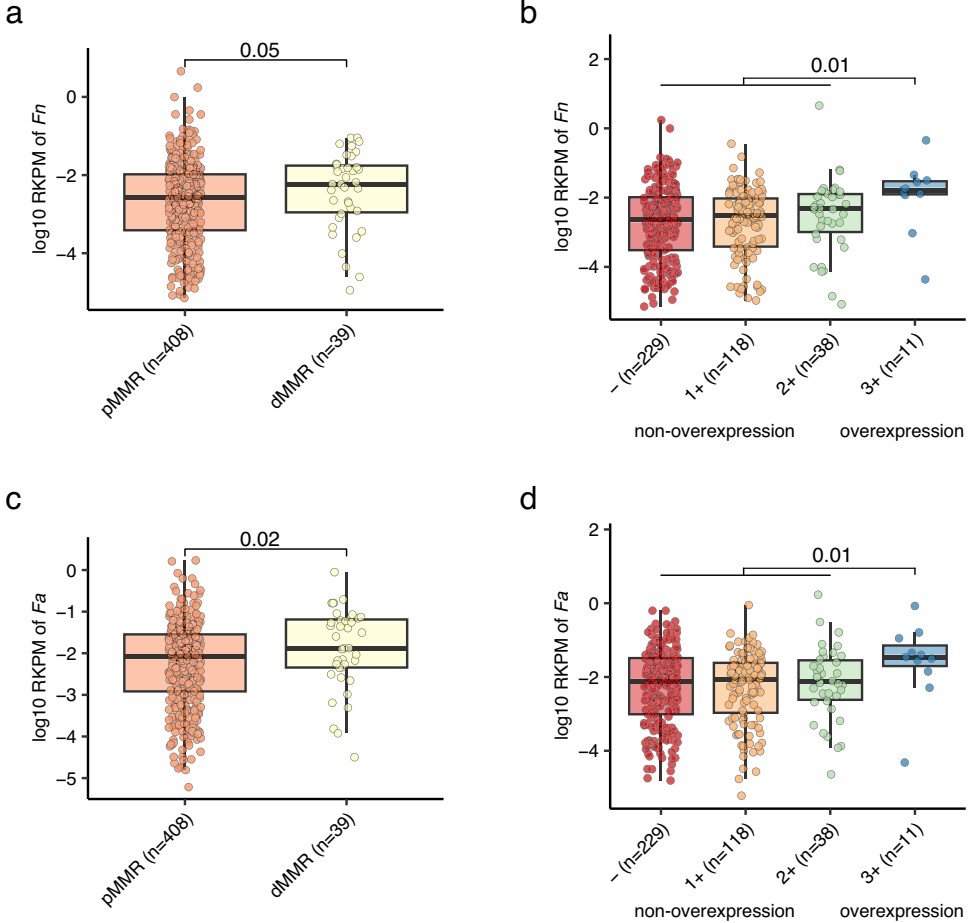

**Fig. 5 | Enrichment of *Fn* and *Fa* in dMMR and HER2 overexpression patients.** The abundances of *Fn* (**a**) and *Fa* (**c**) were higher in dMMR patients compared to pMMR patients. The abundances of *Fn* (**b**) and *Fa* (**d**) were higher in HER2 over-expression patients compared to non-overexpression patients. *Fn* and *Fa* abundances were determined by genome-wide reads mapping to their reference genome (Methods). MMR and HER2 status were determined by immunohistochemistry test. RKPM: reads per kilobase per million reads. *P* values on the top were calculated by two-side Wilcoxon rank-sum test. The boxplot conventions are consistent with the description in Fig. 2.

evaluation to the Guangzhou cohort. The oCRC model demonstrated a high recall rate of 0.7952 when predicting CRC status in Guangzhou oCRC patients, and surprisingly, it outperformed the yCRC model in predicting CRC status in Guangzhou yCRC patients (0.7485 vs. 0.5629, Fig. 6b).

To overcome the limitation of small sample size in the Fudan cohort, we incorporated the publicly available dataset that consisted of 600 CRCs and 662 controls, which included only 72 patients diagnosed under the age of 50[9-16]. The model trained on this expanded dataset predicted CRC status slightly better in Fudan oCRC patients compared to yCRC patients (AUROC = 0.8048 vs. 0.7784, Fig. 6a). Similarly, the model predicted CRC status in the Guangzhou cohort with a slightly higher recall rate in oCRC than yCRC patients (0.8908 vs. 0.8683, Fig. 6b). When we included Guangzhou CRC patients in the training data, the resulting model had similar prediction accuracy for oCRC and yCRC patients in the Fudan cohort (AUROC = 0.7800 and 0.7860, Fig. 6a). The model trained on the public and Fudan datasets also showed similar performance on oCRC and yCRC patients in the Guangzhou cohort (recall rate=0.9044 and 0.9042). In summary, our results suggest that the microbiome-based classifiers can predict CRC status in both old- and young-onset patients with similar accuracy.

Additionally, we trained the random forest models using metagenomic pathway profiles to predict CRC status (Methods). The overall performance of the pathway-based model, as measured by the AUROC and recall rate, was lower than that of the species-based model (Supplementary Fig. 11). This aligns with previous studies reporting that metagenomic pathway-based CRC prediction models tend to exhibit relatively poorer performance compared to species-based models[13,17].

Finally, we replicated our machine learning experiments using another method, least absolute shrinkage, and selection operator (LASSO) logistic regression. Consistent with the findings from the random forest approach, the LASSO models built on species profiles performed better than the pathway profiles (Fig. 6c, d, Supplementary Fig. 11c, d). Importantly, the performance of LASSO models on oCRC and yCRC was similar, suggesting the models' transferability across different age groups.

## Discussion

The CRC microbiome has been extensively studied in old-onset patients, but less is known in young-onset patients. By integrating public data with our own data, we assembled the largest metagenome dataset of young-onset CRC patients to date. Our results revealed that the gut microbiome changes associated with CRC were similar in both old- and young-onset patients in two independent cohorts. We observed the enrichment of key CRC-associated bacteria, including *P. micra*, *P. stomatis*, *C. symbiosum*, and *H. hathewayi*, in both patient groups. CRC-associated virulence factors (*fadA*, *bft*) were enriched in both old- and young-onset CRC compared to their respective controls. Our strain-level analysis reinforced the consistency of CRC-associated

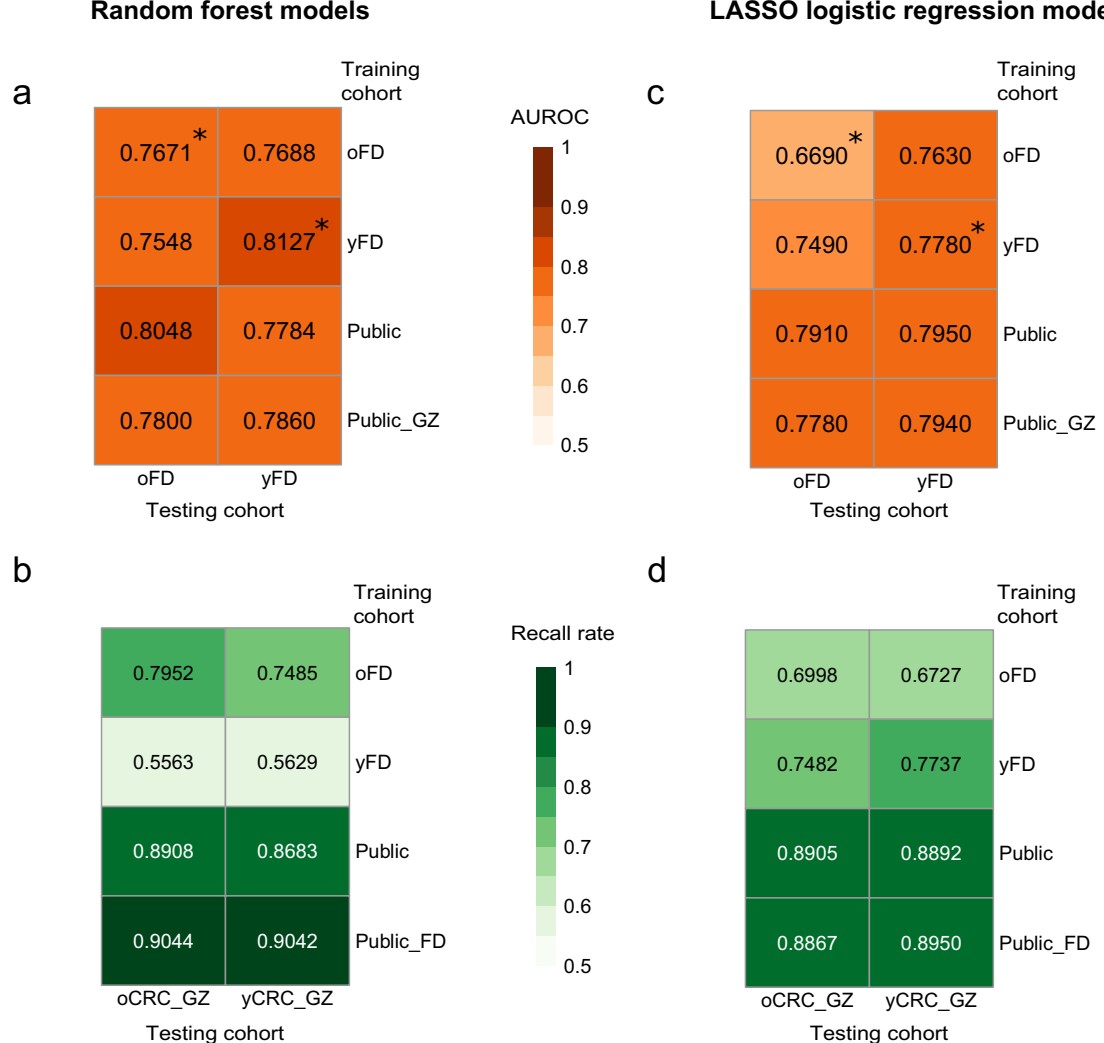

**Fig. 6 | Prediction accuracy of CRC status in old- and young-onset patients.**
Prediction performance on oCRC and yCRC in the Fudan cohort and Guangzhou cohort using models trained on species-level abundances from different datasets. Models were trained on two different methods: random forest and LASSO logistic regression. The numerical values are the area under receiver operator curve (AUROC) for **a** and **c**, and the recall rate for **b** and **d**. Asterisks denote the values averaged over 100 times of 10-fold cross-validation. oFD means the 100 metagenomes of Fudan oCRC and oControl; yFD means the 100 metagenomes of Fudan yCRC and yControl; Public means 1262 public metagenomes; Public_GZ means 1262 public metagenomes plus 460 Guangzhou metagenomes; Public_FD means 1262 public metagenomes plus 200 Fudan metagenomes.

microbial signatures in old- and young-onset CRC. Additionally, the similar prediction accuracy of our microbiome-based models for CRC status in both old- and young-onset patients underscores the consistency of microbial signatures across different age groups.

In contrast to the widespread assumption that intestinal dysbiosis is generally associated with decreased alpha diversity, the gut microbiome in CRC exhibits higher richness than controls[13,14]. Our findings indicated that the increased alpha diversity was not only observed in old-onset patients but also in young-onset patients in both the Guangzhou and Fudan cohorts. However, in the original study of the Fudan cohort, they reported a reduction in alpha diversity in old-onset CRC but an increase in young-onset CRC based on 16S rRNA gene amplicon sequencing[18]. The observed discrepancy may stem from the methodological differences between metagenomic and 16S rRNA gene amplicon data. The robustness of our metagenomic approach in capturing microbial richness highlights the importance of considering sequencing methodologies in microbiome research.

Discrepancy was also found in individual taxa. The Fudan study identified *F. plautii* as an important bacterial species in yCRC, but not in oCRC[18]. We confirmed its enrichment in yCRC patients in our

Guangzhou samples. Whereas the abundance of *F. plautii* was also increased in oCRC in the Guangzhou and Fudan cohorts according to metagenome data. Moreover, the four disease-enriched species (*P. micra, P. stomatis C. symbiosum,* and *H. hathewayi*) were not distinguishable in 16S rRNA data between CRC and controls in the Fudan study[18]. These discrepancies may be accounted by the better resolution of species-level profiling offered by metagenomic data[34]. Notably, *F. nucleatum* was not among the CRC-associated features in the Fudan study[18]. Through genome-wide sequence analysis, we confirmed that *F. nucleatum* can only be detected (with breadth >0.1 and coverage >0.1) in two Fudan samples. The low prevalence of this species did not provide a sufficient sample size for rigorous statistical testing.

Accumulating evidence has shown that insights into microbial strain levels are essential for understanding disease-associated commensal bacteria[5]. Our strain-level analysis of the deep sequencing data found two closely related *Fusobacterium* species *Fn* and *Fa* in the stool metagenome of CRC. This finding aligns with a previous study that highlighted *Fa*'s prevalence over *Fn* in CRC tumor biopsies[29]. Notably, in the final revision of this manuscript, a comprehensive study reported the dominance of a distinct clade of *Fa* in the CRC niche[35]. The

abundance of *fadA* exhibited similar levels in our samples containing either *Fn* or *Fa*, suggesting potential pathogenicity associated with both species. Furthermore, both *Fn* and *Fa* were enriched in patients with dMMR and HER2 overexpression, as opposed to pMMR and HER2 non-overexpression patients, respectively. Given that dMMR patients appear to be susceptible to immunotherapy[36] and *Fn* has been shown to enhance the efficacy of PD-L1 blockage in CRC through both in vivo and in vitro evidence[37], the enrichment of *Fn/Fa* in dMMR CRC raises questions regarding their roles in treatment response. However, further investigations are warranted to understand the interplay between *Fn/Fa* and treatment response, as *Fn* has also been reported to induce resistance to immunotherapy[38]. Taken together, our study demonstrates the dynamics of *Fusobacterium* species in CRC, shedding light on their varying abundance across distinct molecular subtypes.

We found two *B. fragilis* strain clusters in CRC. Interestingly, the abundance and prevalence of enterotoxin gene *bft* was higher in cluster with strains closely related to NCTC9343 than cluster with strain closely related to Q1F2. This example supports existing literature on the complex roles of *B. fragilis* in human health and disease[5]. However, further studies are required to determine the identity of these strains.

Our metagenomic functional analysis revealed an age-independent enrichment of the N-acetylviosamine biosynthesis pathway in the CRC microbiome. Given that N-acetylviosamine is a component of the O-antigen, which is part of the lipopolysaccharide (LPS)[39], our findings align with a recent study indicating LPS enrichment in blood samples from CRC patients[40]. As LPS is a prevalent product in the human gut microbiome, capable of activating Toll-like receptor 4 and inducing immune responses and inflammation[41], our results support the immune and inflammation hypothesis of CRC carcinogenesis[4].

While our study contributes valuable insights into the age-independent changes in the gut microbiome associated with CRC, it is crucial to acknowledge certain limitations. Given the incidence rate of young-onset CRC is about an order of magnitude lower than old-onset CRC[42], the number of young-onset patients in our study was smaller than that of old-onset patients. Despite this limitation, our investigation remains the most extensive analysis of the gut microbiome in young-onset CRC patients to date. Moreover, we cannot definitively establish whether the CRC-enriched microbes are merely associated with the disease or have causal roles in carcinogenesis. Despite age-related changes in the gut microbiome observed in the general population[23], our results unveiled convergent changes in gut microbiome in old- and young-onset CRC patients, regardless of age. This implies that the tumor status may drive the alteration of the microbial ecosystem in the gut, surpassing the age-related changes. Although Mendelian randomization studies have highlighted potential causal links between microbial factors and diseases[43,44], its application in CRC-associated markers remains challenging. One obstacle is the low prevalence of CRC-associated markers in the general population, requiring large sample size to obtain reliable genetic instruments. It is important to note the limitations stemming from the absence of dietary information in our study. Dietary factors play an important role in shaping the gut microbiome. Given the complexity of Chinese cooking culture (characterized by diverse and complex dishes)[45], obtaining precise and reliable dietary information was out of the scope of this study. The absence of dietary information in our study emphasizes the need for future investigations to explore the intricate interplay between diet, convergent changes in the gut microbiome, and their potential roles in promoting carcinogenesis.

Microbiome-based models have been explored and strongly support the promise of non-invasive CRC diagnostics[13,14]. Our study expanded this promise by demonstrating that the model could be applied across a broad age range. Fecal sample has been recognized as an ideal source for non-invasive CRC screening. Currently, the main large-scale implemented CRC screening tests include the fecal immunochemical test (FIT) and the fecal DNA methylation test. However, both methods lack an optimal threshold for young and old populations, as the fecal hemoglobin concentration varies with age and sex[46], and DNA methylation changes are more likely absent in young patients than old patients[47]. The similar prediction accuracy of the gut microbiome-based prediction model shown in our study may facilitate the generalization of CRC screening to all adulthood.

In conclusion, our study highlights the age-independent signatures in the gut microbiome of CRC. The identified microbial patterns emphasize the potential of microbiome-based models for non-invasive CRC diagnostics across a diverse age group. Our findings also demonstrate an example of investigating disease-associated microbial signatures at the strain-level, contributing to a more nuanced understanding of the intricate relationships between the microbiome and CRC.

## Methods
### Study cohorts and data
**Guangzhou cohort.** All patients were recruited from Sun Yat-sen University Cancer Center in Guangzhou in accordance with the study protocol approved by the Ethics Committee of Sun Yat-sen University Cancer Center (B2019-214-X02). Informed consent was obtained from every patient. The inclusion criteria were newly diagnosed pathologically proven locally advanced colon or rectal adenocarcinoma. Patients with any previous tumor history, tumor treatment history, antibiotics, and/or probiotics treatment within 1-month were removed from the study. Clinical data was obtained from regular medical checkups and questionaries. The tumor molecular characteristics (MMR, HER2, and BRAF mutation status) were determined by immunohistochemistry (IHC) test. dMMR was defined as loss expression in any of MSH2, MSH6, MLH1, and PMS2 proteins. HER2 overexpression was defined as strong (3⁺) membranous staining by IHC[48]. A summary of metadata for the recruited patients was given in Supplementary Data 1.

Stool samples were collected by patients following the manufacturer's instructions either at home or in the hospital[49]. Samples were transferred to the laboratory and stored at −80 °C within 7 days and kept thawed until shipped to BGI-Shenzhen. Stool DNA was extracted using MagPure stool DNA KF kit B (no. MD5115-02B). DNA concentrations were estimated using Qubit (Invitrogen). The DNA library was constructed with 200 ng DNA as input. Shotgun metagenomic sequencing was then performed on the BGI-seq platform[50] to generate at least 10 million paired-end reads (length 100 bp) for each sample.

**Fudan cohort.** For the Fudan cohort, sequencing data was downloaded from the NIH National Center for Biotechnology Information Sequence Read Archive (SRA) with BioProject ID PRJNA763023. In total, there were 200 metagenomes consisting of four individual groups (50 samples in each group): old control (oControl), old-onset colorectal cancer (oCRC), young control (yControl), and young-onset colorectal cancer (yCRC). The age cutoff for old and young-onset CRC was 50 years old. Details of the cohort can be found in the original study by Yang et al.[18].

**Public cohort.** The public cohort comprised samples from eight different studies[9–16]. Instead of downloading original sequencing reads, we obtained the processed MetaPhlAn3 taxonomic and HUMAnN3 functional profiles from the supplementary table in Beghini et al.[17]. There were 600 CRC patients and 662 controls. The number of samples in each cohort stratified by age was provided in Supplementary Table 1.

### Metagenomic sequencing data processing
The quality control of sequencing reads was done according to the metapi workflow (https://github.com/ohmeta/metapi/). Sequences

with average quality score below Q30 (0.001% error rate) were removed. Adapter sequences and low-quality tails were trimmed. Only sequences with a length of 70 bp or more were retained. High-quality sequences were then aligned to the human reference genome build hg38 with bowtie2 (v2.4.2)[51]. Sequences mapped to the human genome under *--very-sensitive* mode were removed, and the resulting sequences were used in downstream analysis. Supplementary Data 9 provides a summary of each quality control step. Since 449 out of 460 samples (97.6%) had over 10 million paired-end reads for downstream analysis, we did not subsample the sequences to the lowest sequencing depth to avoid data loss.

To ensure the comparability of our analysis with several large CRC microbiome studies, we calculated the taxonomic and functional abundance profiles using MetaPhlAn3 and HUMAnN3[17]. Default parameters were used, and the smallest relative abundance value was set to $1 \times 10^{-5}$.

We also applied the same quality control and taxonomic analysis to the 200 metagenomes of Fudan cohort.

## Alpha and beta diversity

We assessed the microbiome alpha-diversity using Shannon index and the observed number of species, and beta-diversity using the Bray−Curtis distance. The alpha-diversity was only evaluated on the species abundance profile, while the beta-diversity was evaluated on both the species and pathway abundance profiles. R package *vegan* (version 2.6.4) was used for these calculations.

In our Guangzhou cohort, the Shannon index did not show a correlation with sequencing depth ($rho = -0.027$, $P = 0.56$). We used permutation multivariate analysis of variance (PERMANOVA) based on the Bray−Curtis distance matrix to estimate the variance explained by age, sex (self-report), body mass index (BMI), smoking, tumor location and stage in the Guangzhou cohort. We also used PERMANOVA to evaluate the disease and batch effect when integrating the Guangzhou and Fudan cohorts. For each experiment, the number of permutations was set to 10,000.

## Identification of differential taxa and microbial pathway

To evaluate the correlation between the abundance of each species and age, we used Spearman correlation with and without correction for covariates. As most of the previously reported CRC-associated microbes presented in >15% of the individuals in the Guangzhou cohort, we focused on taxa with a prevalence rate of at least 15% to ensure the power of analysis. We corrected for covariates by first applying the central log-ratio (CLR) transformation to the relative abundance profile[52], followed by fitting a linear regression model with sex, BMI, tumor location and stage, family history of CRC, and smoking as covariates. We then calculated the Spearman correlations between the residuals and age. The obtained Spearman correlation coefficients demonstrated high concordance with those derived from MaAsLin2 (v1.16.0)[53], a widely-used microbiome analysis tool (Pearson correlation coefficients = 0.95 and 0.84, with and without correction for covariates). In addition, unweighted analysis based on the presence and absence of taxa was performed using MaAsLin2, by fitting linear models with and without correction for covariates. The same analyses were applied to microbial pathways.

We used Wilcoxon−Mann−Whitney test to identify microbial taxa and pathways with differential abundance between CRC patients and controls in the Fudan (50 oCRC, 50 oControl, 50 yCRC, and 50 yControl) and Guangzhou cohorts (293 oCRC and 167 yCRC). We conducted four different comparisons: (1) Fudan oCRC versus Fudan oControl, (2) Fudan yCRC versus Fudan yControl, (3) Guangzhou oCRC versus Fudan oControl, and (4) Guangzhou yCRC versus Fudan yControl. We corrected *P* values for multiple hypotheses using Benjamini−Hochberg (FDR) procedure for all taxa in each comparison. This analysis was conducted in R.

## Strain-level analysis of *F. nucleatum*, *B. fragilis*, and *E. coli*

We used two distinct methods, StrainPhlAn3[17] and inStrain (v1.8.0)[28], to explore the strain-level diversity of three well-known CRC-associated bacteria: *F. nucleatum* (*Fn*), *B. fragilis* (*Bf*), and *E. coli* (*Ec*). StrainphlAn3 relies on marker genes, while inStrain considers the genome-wide sequences. Our analysis focused on 460 Guangzhou samples, which had higher sequencing depth than the Fudan and public samples.

For StrainPhlAn3, default parameters were applied, requiring a minimum of 20 marker genes per sample, each marker gene found in at least 80% of samples. *Fn* was found in 63 (14%) samples based on 36 marker genes. *Bf* was found in 334 (73%) samples based on 46 marker genes. *Ec* was found in 317 (69%) samples based on 24 marker genes.

For inStrain analysis, we built a comprehensive reference genome from de novo assembly and NCBI RefSeq genomes. Using the metapi pipeline, which employed megahit (v1.2.9)[54] for genome assembly, metabat2[55] for binning, and bowtie2 for alignment, we obtained 37,441 metagenome-assembled genomes (MAGs) with a minimum length of 200 kb. We then used GTDB-Tk (v2.1.0)[56] to annotate these MAGs referring to GTDB release207[57] with default parameters. For *Fn*, only 6 samples had MAGs annotated to *Fusobacterium nucleatum_J* (RefSeq assembly GCF_008633215.1, NCBI strain identifier 13-08-02), while 11 samples had MAGs annotated to *Fusobacterium animalis* (*Fa*, RefSeq assembly GCF_000158275.2, strain identifier 7_1). *Fa* was also known as *Fn* subspecies *animalis*. For *Bf*, 261 samples had MAGs annotated to RefSeq assembly GCF_000025985.1 (NCBI strain NCTC9343), and 60 samples had MAGs annotated to RefSeq assembly GCF_002849695.1 (NCBI strain Q1F2). For *Ec*, 285 samples had MAGs annotated to RefSeq assembly GCF_003697165.2 (NCBI strain ATCC 11775). As a sanity check, samples with MAGs annotated to *Fn*, *Bf*, and *Ec* strains were recovered in the corresponding StrainPhlAn3 analysis. We combined 6356 high-quality MAGs (>90% completeness and <5% contamination, according to the definition in Bowers et al.[58]) and 5 RefSeq genomes (mentioned above) together and used dRep (v3.2.0)[59] to select a unique set of reference genomes. After dereplication, 3084 genomes remained, including the 5 RefSeq genomes. Subsequently, reads from 460 Guangzhou samples were aligned to the dereplicated genomes using bowtie2, and the inStrain analysis was applied to the 5 RefSeq genomes.

Given the varied abundances of different taxa, we used taxon-specific parameters for the interpretation of inStrain results. For *Fn* and *Fa*, we used breadth >0.1 and coverage >0.1 (meaning that at least 10% of the reference genome was mapped and the average genome-wide mapping frequency is at least 0.1); for *Bf*, breadth >0.5 and coverage >1.0; for *Ec*, breadth >0.1 and coverage >0.2. These criteria retained 50 samples for *Fn* (including 5 with MAGs annotated to RefSeq assembly GCF_008633215.1 and 22 supported by StrainPhlAn3) and 84 samples for *Fa* (including 11 with MAGs annotated to RefSeq assembly GCF_000158275.2). For *Ec*, 328 samples met the criteria, with 299 (91%) supported by StrainPhlAn3.

For *Bf*, 275 samples remained for strain NCTC9343 (271 samples supported by StrainPhlAn3) and 67 samples for strain Q1F2 (66 samples supported by StrainPhlAn3). Of note, strain Q1F2 was not covered in StrainPhlAn3 due to the absence of marker genes for this taxon. We inspected the gene marker-based phylogenetic tree of *Bf*. There were two distinct clusters, one was predominantly composed of samples with MAGs annotated to strain NCTC9343, and the other was dominated by samples with MAGs annotated to strain Q1F2. Consequently, we manually divided the 334 StrainPhlAn3 samples into two clusters. Cluster 1 (related to NCTC9343) comprised 267 samples, all supported by inStrain analysis. Cluster 2 (related to Q1F2) consisted of 67 samples, with 61 supported by inStrain analysis.

## Known CRC-associated microbial taxa

Known CRC-associated microbial taxa were obtained from the gutMDisorder[24] (http://bio-annotation.cn/gutMDisorder) database

with parameters: *Species=Human, Condition=Colorectal Neoplasms (distal cancer), Sequencing Technology=Whole metagenomic sequencing* and *Taxonomy Rank=species*. The resulting list of 148 records covering 118 species was downloaded in March 2023 and shown in Supplementary Data 10.

### CRC-associated genes
We obtained reference sequences of CRC-associated genes from Wirbel et al.[14]. Specifically, representative sequences for *fadA*, *bft*, the *pks* genomic island, and the *bai* operon were identified from integrated gene catalog (IGC)[60] using the gene-specific Hidden Markova Model. Subsequently, metagenomic sequencing reads were mapped to the IGC using bowtie2, and the read count for each gene was calculated. The number of mapped reads for each gene was then normalized by dividing it by the total number of mapped reads in each sample. The resulting number can be interpreted as the number of reads per million mapped reads (RPM). The abundances of *fadA*, *bft*, the *pks* and *bai* were the sum of all their member genes. For the *cutC* gene, we used the output from Humman3 calculations for the gene family. The abundance of the *cutC* gene was determined as the sum of its member gene clusters.

### Microbiome-based classification
We evaluated the microbiome-based classification capabilities for oCRC and yCRC using two widely adopted algorithms: random forest and least absolute shrinkage and selection operator (LASSO) logistic regression. The random forest algorithm, known for its superior performance among various machine learning tools[61], has been successfully employed in previous large-scale CRC microbiome studies[13,17]. The LASSO logistic regression has been used in a comprehensive meta-analysis study on CRC microbiome[14]. The assessment was conducted on two types of microbiome quantitative profiles: species and pathway-level relative abundances calculated by MetaPhlAn3 and HUMAnN3. To enhance the generalizability of our models, we filtered out features not presented in all studied cohorts. This resulted in a dataset comprising 199 taxa and 401 pathways that were consistently found in at least one sample in each of the Guangzhou, Fudan, and public cohorts.

In the random forest task, we used the algorithm implemented in R package *mlr3* (v0.17.2). Each training iteration consisted of an ensemble of 10,000 estimator trees, with the number of features per tree set to the square root of the total number of features. The impurity score was determined by Shannon entropy ('*gini*') to evaluate the quality of tree growing. Other parameters included no-maximum depth for the trees and one sample as the minimum amount for the leaf node. To estimate the within-dataset prediction capability, 10-fold cross-validation was employed for Fudan oCRC and yCRC, and this process was repeated 100 times. The reported results represent an average over 100 validation folds. For cross-study prediction, the model was trained once on all the training samples and applied to the testing samples. These experiments were conducted using the output values from MetaPhlAn3 and HUMAnN3, as the random forest algorithm is robust to data normalization.

In the LASSO logistic regression task, the relative abundances were $\log_{10}$-transformed and standardized as z-scores. Zeros in the species and pathway abundance table were replaced by a pseudo-count of $1 \times 10^{-6}$ and $1 \times 10^{-8}$ before the $\log_{10}$-transformation. During the training phase, a 10-fold cross-validation strategy was employed to tune the lambda parameter (regulation strength). The lambda parameter for each model was selected to maximize the area under the precision-recall curve. We repeated the ten-fold cross-validation 100 times for models trained exclusively on Fudan data and 10 times for models trained on public data (with or without Fudan and Guangzhou data). In the prediction phase, the prediction evaluation was averaged across all models. For models trained on Fudan data, a nested feature

selection step was implemented, and the models with the best performance were reported. All these experiments were conducted in R with package *SIAMCAT* (v2.6.0)[62].

### Statistical analysis
To access the correlation between age and metadata, Guangzhou CRC patients were grouped into six age categories: [20,30], [30,40], [40,50], [50,60], [60,70] and ≥70. Subsequently, the distribution of sex, tumor stage, and location, family history of CRC, smoking status, and tumor molecular characteristics (MMR, MSH2, MSH6, MLH1, PMS2, HER2, and BRAF status) across these age groups was examined using the chi-square test. The Fisher test was employed if any count was <5. Furthermore, patients were divided into two groups based on age, specifically, old-onset (age ≥ 50) and young-onset (age < 50), and the same tests were repeated. For body max index (BMI), the means in oCRC and yCRC were compared using the Wilcoxon–Mann–Whitney test. The association between BMI and age in all patients was measured using Pearson correlation. The R functions *chisq.test()*, *fisher.test()*, *wilcox.test()* and *cor.test()* were used for these analyses.

We employed MaAsLin2 (v1.16.0) to examine associations between microbial markers and tumor characteristics, including tumor stage, location, MMR, HER2, and BRAF mutation status. For each clinical factor, two linear models were fitted: one without adjusting for any covariate and the other adjusted for age, sex, tumor stage and location, family history of CRC. Both weighted and unweighted analyses were performed. In the weighted analysis, relative abundance underwent CLR transformed. For the unweighted analysis, the presence and absence matrix was derived from the relative abundance matrix, with a threshold of zero. Benjamini–Hochberg method was used for multiple testing adjustment. Trend analysis was conducted using ANCOMBC2 (v2.4.0)[63] to assess the monotonic relationship between microbial markers and tumor stage and location. To reduce the burden of multiple tests, we only applied these analyses to 18 taxa which were highlighted by our cohort (Fig. 2, Supplementary Figs. 1–3) and the well-known gene markers (*fadA*, *bft*, *pks*, and *bai*).

### Reporting summary
Further information on research design is available in the Nature Portfolio Reporting Summary linked to this article.

## Data availability
The data generated in this study and data obtained from Yang et al.[18] have been deposited into CNGB Sequence Archive (CNSA)[64] of China National GeneBank DataBase (CNGBdb)[65] with accession number CNP0004314. This archive contains the fastq sequences, de novo assembly, and binning results. A copy of data has also been deposited into the Genome Sequence Archive (GSA) with submission number HRA004617. The metagenomic data of Yang et al. can be found by PRJNA763023.

## Code availability
No unique software or computational code was created for this study. The microbiome data and associated codes used in microbiome-based classification tasks are available on GitHub repository https://github.com/Owen-haha/CRCmicrobiome.

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

## Acknowledgements

This study was supported by the National Natural Science Foundation of China [82073159, 82002467], the Natural Science Foundation of Guangdong [2022A1515012296], Beijing Xisike Clinical Oncology Research Foundation [Y-tongshu2021/ms-0175], Guangdong Genomics Data Center [2021B1212100001], Guangdong Basic and Applied Basic Research Foundation (2020A1515110544) and Cancer Innovative Research Program of Sun Yat-sen University Cancer Center [CIRP-SYSUCC-0032]. We thank all the participating patients for their generous and volunteer support of this study. We thank the Fudan research team for their efforts to make the data publicly available. We thank the China National GeneBank (CNGB) Shenzhen for DNA extraction, library construction, and sequencing. We gratefully thank our BGI colleagues Jie Zhu, Yanmei Ju, Weiting Liang, Liu Tian, Xiaoqian Lin, and Tongyuan Hu for their technical support in bioinformatics.

## Author contributions

Y.Q. and P.D. designed the study. W.M., Y.C. and J.Y. collected the samples. X.T. acquired the data. Y.Q., X.T. and W.M. analyzed the data. Y.Z., Z.J. and H.Z. helped with the microbiome data analysis. Y.Q. drafted the manuscript. X.T., W.M., Y.Z., T.Z. and H.Z. critically read the manuscript. Y.Z., K.H., S.Z., X.J., J.W., H.Y., X.X., H.Z., L.X. and P.D. supervised the study. All authors read the manuscript and gave final approval of the version to be published.

## Competing interests

All authors declare no competing interests.
