## [Peer Review File · Nature Communications]

REVIEWER EXPERTISE

Reviewer #1. Microbiome / Metagenomics / Bioinformatics.

Reviewer #2. CRC /Microbiome / Clinical,

Reviewer #2. CRC / Machine learning / Statistics.

REVIEWER COMMENTS

Reviewer #1 (Remarks to the Author):

Thank you for inviting me to review the work by Qin et al, where the authors aim to describe the taxonomic and imputed functional characteristics of the fecal microbiome in patients with old- and young-onset colorectal cancer.

This study contributes to the existing body of literature that identifies associations between oral and gut taxa and the presence of colorectal lesions. Importantly, it attempts to compare the composition of the fecal microbiome in young-onset colorectal cancer patients with that of old-onset colorectal cancer patients, using a substantial number of participants. However, there are several aspects of this work that could be improved to provide novel and informative contributions to the field.

Despite its title, this study does not evaluate "changes" in the human gut microbiome of patients with old- and young-onset colorectal cancer, as it is not a longitudinal study. On the contrary, the presented analysis focuses on differences between these two distinct cohorts and respective control groups.

The evaluation of alpha diversity should include an analysis based on observed species. While using indices such as Shannon is important for assessing the structure of diversity, the sheer number of species also provides informative measurements. Previous studies have described the impact of age and sex on the diversity and composition of the gut microbiome. Therefore, it would be highly beneficial to assess multiple variables using multivariate multiple regression.

However, the major limitation of the current study is the absence of taxonomic and phylogenetic analyses. This is likely due to the chosen methodology used to characterize the composition of the fecal microbiome. The presence of oral bacteria in fecal samples from colorectal cancer patients has been documented for over a decade. However, the field has progressed to identify potential mechanisms through which these bacteria may contribute to the onset and progression of colorectal cancer and associated metastases, as outlined by the authors in the introduction. It is been suggested that specific subspecies of *Fusobacterium nucleatum*, such as subspecies *animalis* but not subspecies *nucleatum*, are more prevalent in colorectal cancer tissues. Furthermore, studies have demonstrated that these strains differ from those found in the oral cavity. Given the large volume of sequencing reads available (70 million paired-end reads per sample), the authors have an excellent opportunity to explore the diversity within the identified species in these cohorts. This analysis can help elucidate differences between young-onset, old-onset, and control cohorts at a high resolution in terms of taxonomy and phylogeny. I anticipate that such an approach would yield novel and informative data for the field. Additionally, it would provide further insights into genetic determinants observed within the species of interest and inform the imputed functional analyses.

Reviewer #2 (Remarks to the Author):

Major revision

The authors aimed to investigate the microbiome characteristics in early-onset colorectal cancer (CRC) and compare them to those in later-onset colorectal cancer. The research topic is very interesting and scientifically important. I have several comments on this manuscript as follows.

1) This may be a minor point, but I think yCRC and oCRC are not common abbreviations. I recommend spelling out young-onset CRC and old-onset CRC. This may be my preference, but I feel that early-onset CRC and later-onset CRC are more common terminology. To improve the readability, I recommend using these terms.

2) If I don't miss something, I cannot find tables showing clinical and tumor characteristics in all cases, early-onset cases, and later-onset cases in each cohort (e.g., Guangzhou and Fudan cohorts). This is very important information. The authors should present.

3) In addition, I expect that there are some differences in characteristics between early-onset and later-onset cases. How did the authors adjust for such differences? The authors should explicitly show whether there are differences in clinical and tumor characteristics, and adjust for those differences in age-analyses. This is a very important point. For example, if early-onset cases have

more advanced stages, observed differences by age groups can be due to advanced stages, not age. Please carefully consider this point.

4) Other important characteristics are tumor characteristics (e.g., microsatellite instability, BRAF, KRAS mutations). If the authors consider these tumor molecular characteristics into account, that would be great. If it is not possible, the authors could mention these as a limitation or a next step for this study.

5) Accumulating evidence indicates that bacterial toxins are involved in colorectal cancer carcinogenesis and progression. For example, pks+ *E. coli* has been reported to play an important role in colorectal carcinogenesis. If the authors could include analyses of bacterial toxins, those would be super interesting.

6) In the discussion, the authors could discuss why reported CRC-associated microbiome, such as *Fusobacterium nucleatum* and *Bacteroides fragilis*, did not show a strong association in this study. In addition, enterotoxigenic *Bacteroides fragili* is included in *Bacteroides fragilis*. Ideally, the authors should separate enterotoxigenic *Bacteroides fragili* from *Bacteroides fragilis*. If it is not possible, the authors could discuss this point in the discussion.

Minor comments

1) "We found no correlation between age and alpha diversity, defined as the Shannon index, at the species level ($\rho=-0.04$, $P=0.42$, Figure 1b). Similarly, there was no correlation between age and the first two coordinates of the principal coordinate analysis (PCoA) (Figure 1c, d & e)."

I think these labels for Figure 1 were wrong. The first sentence corresponds to Figure 1c? The second sentence corresponds to Figure 1d and 1e? Please check.

2) In Figure 3 and other Figures, darker colors reflect lower AUROCs or rates. This is not intuitive. In my opinion, darker colors should reflect higher AUROCs or rates.

3) The tiles and footnotes for Supplementary Tables are not shown correctly in the PDF file. Please correct them.

Reviewer #3 (Remarks to the Author):

1. The language of this manuscript should be improved to avoid grammatical errors and inappropriate descriptions. For example, "While the genetic background of world population is unlikely changed over the last several decades" in line 57, "comparably high accuracy" in line 85, and "The CRC microbiome has been heavily studied in old-onset patients" in line 276. The manuscript's

wording could be made easier to read by avoiding confusing phrases, e.g., "the only publicly available yCRC dataset", "all other publicly available metagenomic CRC datasets", and "the so far only published yCRC cohort" in line 139.

2. The authors controlled for BMI, sex, smoking, tumor location and stage, and family history of CRC in testing the correlation between age and specific taxa. As diet strongly influences the microbiome, it is essential for the authors to consider the effects of diet in their study.

3. The title "Comparable prediction accuracy of CRC status in old- and young-onset patients" is confusing. It is unclear what "comparable" refers to in this context. Is it meant to convey similarity or capability for comparison? A more precise title is recommended.

4. In Figure 3, it would be beneficial to show all the testing results, such as the results of the model trained on oCRC_GZ/yCRCGZ in oCRC_FD/yCRC_FD. Additionally, the authors should ensure consistency by showing both AUROC and recall rate in a testing cohort to make them comparable. Currently, Figure 3 displays AUROC in oCRC_FD/yCRC_FD, but recall rate in oCRC_GZ/yCRCGZ, which may be improved for clarity.

5. While the authors cited previous studies in methods to claim that random forest performs better than other methods, it is important to note that the performance of different methods can vary with different datasets and sample sizes. Considering the authors' collection of a large dataset, it is worth exploring comparisons with other methods using their own data.

6. The yCRC_FD trained model's poorer performance in oCRC_FD than in oCRC_GZ/yCRC_GZ raises concerns about potential overfitting problems. Further investigation and clarification on this matter are necessary.

7. There appears to be a sudden transition from displaying prediction models to pathway profiles. To improve the flow of the manuscript, the authors should ensure a smoother transition between these topics.

8. In the discussion of microbiome-based classification, the authors claimed that "Our study highlights the potential advantages of microbiome-based CRC prediction for screening." However, most of the models in this manuscript did not outperform the previous studies cited in the methods (Thomas et al., 2019; Beghini et al., 2021). This claim may be too strong based on the results and should be appropriately revised to accurately reflect the findings.

REVIEWER COMMENTS

Reviewer #1 (Remarks to the Author):

Thank you for inviting me to review the work by Qin et al, where the authors aim to describe the taxonomic and imputed functional characteristics of the fecal microbiome in patients with old- and young-onset colorectal cancer.

This study contributes to the existing body of literature that identifies associations between oral and gut taxa and the presence of colorectal lesions. Importantly, it attempts to compare the composition of the fecal microbiome in young-onset colorectal cancer patients with that of old-onset colorectal cancer patients, using a substantial number of participants. However, there are several aspects of this work that could be improved to provide novel and informative contributions to the field.

Response: We thank the reviewer for the kind comment, we have addressed their major and minor concerns below.

Despite its title, this study does not evaluate "changes" in the human gut microbiome of patients with old- and young-onset colorectal cancer, as it is not a longitudinal study. On the contrary, the presented analysis focuses on differences between these two distinct cohorts and respective control groups.

Response: We agree with the reviewer and have replaced the “changes” to “signatures”. The revised title is “Consistent signatures of human gut microbiome in young- and old-onset colorectal patients”.

The evaluation of alpha diversity should include an analysis based on observed species. While using indices such as Shannon is important for assessing the structure of diversity, the sheer number of species also provides informative measurements. Previous studies have described the impact of age and sex on the diversity and composition of the gut microbiome. Therefore, it would be highly beneficial to assess multiple variables using multivariate multiple regression.

Response: We have added the number of observed species to our results, the new **Figure 1e**. To account for the effect of confounding factors, we fitted a linear regression model with sex, BMI, tumor stage and location, family history of CRC and smoking as covariates. We then calculated the Spearman correlations between the residuals and age. Similar to Shannon index, we did not observe significant association ($P>0.1$) between richness (the number of species) and patient age, with and without the adjustment for confounders. We have updated the text accordingly at lines 112-113.

However, the major limitation of the current study is the absence of taxonomic and phylogenetic analyses. This is likely due to the chosen methodology used to characterize the composition of the fecal microbiome. The presence of oral bacteria in fecal samples from colorectal cancer patients has been documented for over a decade. However, the field has progressed to identify potential

mechanisms through which these bacteria may contribute to the onset and progression of colorectal cancer and associated metastases, as outlined by the authors in the introduction. It is being suggested that specific subspecies of *Fusobacterium nucleatum*, such as subspecies *animalis* but not subspecies *nucleatum*, are more prevalent in colorectal cancer tissues. Furthermore, studies have demonstrated that these strains differ from those found in the oral cavity. Given the large volume of sequencing reads available (70 million paired-end reads per sample), the authors have an excellent opportunity to explore the diversity within the identified species in these cohorts. This analysis can help elucidate differences between young-onset, old-onset, and control cohorts at a high resolution in terms of taxonomy and phylogeny. I anticipate that such an approach would yield novel and informative data for the field. Additionally, it would provide further insights into genetic determinants observed within the species of interest and inform the imputed functional analyses.

Response: We thank the reviewer for their recognition of our efforts to investigate the CRC microbiome with a relatively large number of deep metagenomes. To obtain an in-depth taxonomic and phylogenetic picture from our shotgun metagenomic data, we performed *de novo* assembly and binning to generate metagenome-assembled genomes (MAGs), followed by taxonomic annotation and strain-level analyses on three well-known CRC-associated species: *Fusobacterium nucleatum*, *Bacteroidetes fragilis* and *Escherichia coli*. We used StrainPhlAn3 to construct the phylogenetic tree and used inStrain to examine the genome-wide sequence diversity.

For *F. nucleatum*, our analysis identified both *F. nucleatum* (*Fn*) and *F. nucleatum* subspe. *animalis* (also known as *F. animalis*, *Fa*) in our CRC cohort (new **Figure S4f**). In line with Younginger *et al.* 2023, *Cell Rep Med*, the prevalence and abundance of *Fa* were higher than *Fn* in the stool microbiome of our Chinese CRC patients.

For *B. fragilis*, we revealed two distinct phylogenetic clusters (new **Figure S6**). Cluster 1 was predominantly composed of samples with MAGs annotated to strain NCTC9343 (average nucleotide identity (ANI) >95%), and cluster 2 was dominated by samples with MAGs annotated to strain Q1F2. More importantly, both abundance and prevalence of the *bft* gene were higher in cluster 1 compared to cluster 2 (new **Figure 3c**, P<0.01).

For *E. coli*, our analysis identified only one strain cluster (ANI >95% to strain ATCC 11775), with no significant difference in prevalence between oCRC and yCRC (new **Figure S7**).

We have included a dedicated section (lines 208 to 252) to the **Results** to describe these findings, as well as a detailed method of these analyses to the **Methods** (lines 544-594).

To facilitate the CRC microbiome research, we have uploaded the *de novo* assembly and binning results to CNGB (<https://db.cngb.org/search/project/CNP0004314/>). The data will be released once the paper is published.

Reviewer #2 (Remarks to the Author):

Major revision

The authors aimed to investigate the microbiome characteristics in early-onset colorectal cancer (CRC) and compare them to those in later-onset colorectal cancer. The research topic is very interesting and scientifically important. I have several comments on this manuscript as follows.

Response: We thank the reviewer for their kind comments and appreciation of our work. We have addressed all their comments below, hopefully in a satisfactory way.

1) This may be a minor point, but I think yCRC and oCRC are not common abbreviations. I recommend spelling out young-onset CRC and old-onset CRC. This may be my preference, but I feel that early-onset CRC and later-onset CRC are more common terminology. To improve the readability, I recommend using these terms.

Response: In reference to a prior study by Yang *et al.* (2021) in *Nature Communications*, the terms yCRC and oCRC were used. For consistency, we adopted these abbreviations in our manuscript. However, we acknowledge the reviewer's suggestion to enhance readability by using the full terms. We have implemented this change in our manuscript whenever we think the full names are more reader friendly. To facilitate our communications, we used early-onset CRC and later-onset CRC in our response to reviewer #2 for clarity.

2) If I don't miss something, I cannot find tables showing clinical and tumor characteristics in all cases, early-onset cases, and later-onset cases in each cohort (e.g., Guangzhou and Fudan cohorts). This is very important information. The authors should present.

Response: In our original submission, we provided summary information for clinical and tumor characteristics of the Guangzhou cohort in **Table S8**. To enhance clarity, we have now included the summary information for both early-onset and later-onset cases (now in revised **Table S8**). We also updated the methods sections (lines 657-669) to describe how we calculated the P values in this table.

For the Fudan cohort, the clinical and tumor characteristics have been previously published in Supplementary Table 2 in Yang *et al.*

3) In addition, I expect that there are some differences in characteristics between early-onset and later-onset cases. How did the authors adjust for such differences? The authors should explicitly show whether there are differences in clinical and tumor characteristics, and adjust for those differences in age-analyses. This is a very important point. For example, if early-onset cases have more advanced stages, observed differences by age groups can be due to advanced stages, not age. Please carefully consider this point.

Response: We acknowledge the importance of accounting for phenotypic differences between early-onset and later-onset cases in microbiome analysis. In our original submission, we presented the age analysis results in **Table S2 & S5**, with and without the adjustment for confounders. To

account for the effects of confounding factors, we applied the central log-ratio transformation (CLR) to the relative abundance profile, followed by fitting a linear regression model with BMI, sex, tumor stage and location as covariates. We then calculated the Spearman correlations between the residuals and age.

In the revised **Table S8**, we have included analysis results of these clinical and tumor characteristics between early-onset and later-onset CRC. Notably, only BMI showed a weak positive association with age (Pearson correlation=0.12, $P=0.01$), with later-onset CRC demonstrating a higher BMI than early-onset CRC (mean BMI 22.23 vs. 22.37, $P=0.008$). We have updated text at lines 104-105.

4) Other important characteristics are tumor characteristics (e.g., microsatellite instability, BRAF, KRAS mutations). If the authors consider these tumor molecular characteristics into account, that would be great. If it is not possible, the authors could mention these as a limitation or a next step for this study.

Response: Indeed, our study was designed to encompass important molecular characteristics, including mismatch repair system (pMMR vs dMMR) and status of MSH2, MSH6, MLH, PMS2, HER2 and BRAF. The summary information was given in **Table S8**. Notably, only the mismatch repair status showed a different distribution between early-onset and later-onset CRC (6.38% vs. 12.73%, $P=0.03$). However, due to the limited sample size in each category, we were unable to obtain any statistically meaningful result for the associations between these molecular characteristics and microbiome features. It would be our future research aim when we have a sufficiently large sample size. We recognize this as a limitation and have highlighted it in the updated discussion (lines 416-420).

5) Accumulating evidence indicates that bacterial toxins are involved in colorectal cancer carcinogenesis and progression. For example, pks+ *E. coli* has been reported to play an important role in colorectal carcinogenesis. If the authors could include analyses of bacterial toxins, those would be super interesting.

Response: We appreciate the insightful suggestion. In response, we have added an analysis of virulence factors and toxins (*fadA*, *bft*, the *pks* island and the *bai* operon) in the revised manuscript. In line with previous findings, our results show an enrichment of virulence factors and toxins in early-onset and later-onset CRC compared to their respective controls. A dedicated paragraph (lines 287 to 309) and a main figure (new **Figure 3**) have been added to the manuscript to describe our findings.

6) In the discussion, the authors could discuss why reported CRC-associated microbiome, such as *Fusobacterium nucleatum* and *Bacteroides fragilis*, did not show a strong association in this study. In addition, enterotoxigenic *Bacteroides fragilis* is included in *Bacteroides fragilis*. Ideally, the authors should separate enterotoxigenic *Bacteroides fragilis* from *Bacteroides fragilis*. If it is not possible, the authors could discuss this point in the discussion.

Response: We appreciate the reviewer's keen observation, and we have conducted a comprehensive strain-level analysis. In our cohort, two prevalent *Bacteroides fragilis* strain

clusters (ANI >95% to reference genomes of NCTC 9343 and Q1F2, respectively) were identified. Notably, we found that the abundance and prevalence of the *bft* gene were significantly differentiated between the two clusters ($P < 0.02$). These findings have been incorporated into the revised manuscript (lines 234-244 & 295-297 and new **Figure 3**).

Regarding *Fusobacterium nucleatum* (*Fn*), its prevalence was low in the Fudan cohort, with only 9 CRC samples (5 early-onset and 4 later-onset samples) showing non-zero *Fn* abundance according to MetaPhlan3 calculations. To validate this, we conducted genome-wide sequence mapping to the *Fn* reference genome (RefSeq assembly GCF_008633215.1) and found similar results that only two samples (2 early-onset and 0 later-onset) exhibited reasonable mapping (breadth > 0.1 and coverage > 0.1). Consequently, the low prevalence of *Fn* hampered our ability to robustly identify its association with CRC. This explanation has been added to the manuscript (lines 386-389).

Minor comments

1) “We found no correlation between age and alpha diversity, defined as the Shannon index, at the species level ($\rho = -0.04$, $P = 0.42$, Figure 1b). Similarly, there was no correlation between age and the first two coordinates of the principal coordinate analysis (PCoA) (Figure 1c, d & e).” I think these labels for Figure 1 were wrong. The first sentence corresponds to Figure 1c? The second sentence corresponds to Figure 1d and 1e? Please check.

Response: We thank the reviewer for pointing out this error. We have corrected this error at lines 112-116.

2) In Figure 3 and other Figures, darker colors reflect lower AUROCs or rates. This is not intuitive. In my opinion, darker colors should reflect higher AUROCs or rates.

Response: We have updated the figures (now **Figure 4** and **Figure S11**) according to reviewer’s suggestions.

3) The tiles and footnotes for Supplementary Tables are not shown correctly in the PDF file. Please correct them.

Response: We have double checked and made it clear in the new PDF file.

Reviewer #3 (Remarks to the Author):

1. The language of this manuscript should be improved to avoid grammatical errors and inappropriate descriptions. For example, "While the genetic background of world population is unlikely changed over the last several decades" in line 57, "comparably high accuracy" in line 85, and "The CRC microbiome has been heavily studied in old-onset patients" in line 376. The manuscript's wording could be made easier to read by avoiding confusing phrases, e.g., "the only publicly available yCRC dataset", "all other publicly available metagenomic CRC datasets", and "the so far only published yCRC cohort" in line 139.

Response: We appreciate the reviewer's feedback. We have addressed these concerns by revising and simplifying the relevant sentences throughout the manuscript. For instance, in lines 56-58, 89, 142 and 356-357. Additionally, we have conducted a thorough review of the entire manuscript, editing unclear words and sentences for improved readability.

2. The authors controlled for BMI, sex, smoking, tumor location and stage, and family history of CRC in testing the correlation between age and specific taxa. As diet strongly influences the microbiome, it is essential for the authors to consider the effects of diet in their study.

Response: We acknowledge the importance of considering the influence of diet on the microbiome. Unfortunately, dietary information was not collected in our study due to the inherent challenges in accurately quantifying food and nutrition intake in our cohort. Given the complexity of Chinese cooking culture (characterized by diverse and complex dishes), obtaining precise and reliable dietary information was unfeasible within the scope of our study. This limitation has been explicitly addressed in the updated discussion (lines 425-431).

3. The title "Comparable prediction accuracy of CRC status in old- and young-onset patients" is confusing. It is unclear what "comparable" refers to in this context. Is it meant to convey similarity or capability for comparison? A more precise title is recommended.

Response: We thank the reviewers for pointing this out. We have revised the title to "Similar prediction accuracy of CRC status in old- and young-onset patients" (line 311). Additionally, we have carefully checked through the manuscript to ensure that other potentially unclear words are edited for clarity.

4. In Figure 3, it would be beneficial to show all the testing results, such as the results of the model trained on oCRC_GZ/yCRC_GZ in oCRC_FD/yCRC_FD. Additionally, the authors should ensure consistency by showing both AUROC and recall rate in a testing cohort to make them comparable. Currently, Figure 3 displays AUROC in oCRC_FD/yCRC_FD, but recall rate in oCRC_GZ/yCRC_GZ, which may be improved for clarity.

Response: Our objective was to compare model performance in oCRC and yCRC. Therefore, models trained on different cohorts were used to predict CRC status in cohorts with both oCRC and yCRC. In the Fudan cohort, which included oCRC, yCRC, and respective controls, we used AUROC to evaluate model performance. However, in the Guangzhou cohort, where only oCRC

and yCRC were present without controls, AUROC calculation was not feasible. Instead, we calculated recall rate to compare the model performance in oCRC and yCRC. To distinguish these metrics, we intentionally used different colors and abbreviations (oFD, yFD, oCRC_GZ, yCRC_GZ) in the revised **Figure 4 & S11**. This approach allows for a clear representation of the different evaluations in each cohort.

5. While the authors cited previous studies in methods to claim that random forest performs better than other methods, it is important to note that the performance of different methods can vary with different datasets and sample sizes. Considering the authors' collection of a large dataset, it is worth exploring comparisons with other methods using their own data.

Response: We appreciate the reviewer's suggestion. To provide a more comprehensive comparison, we have applied least absolute shrinkage and selection operator (LASSO) logistic regression to our machine learning analysis. The LASSO models demonstrated similar performance to random forest models in both oCRC and yCRC. These findings, along with additional details, have been incorporated into the revised manuscript (lines 345-350 & 620-655, and **Figure 4 & S11**).

To facilitate access to the microbiome-based CRC prediction analysis, we have created a GitHub repository (<https://github.com/Owen-haha/CRCmicrobiome>), which includes both the data and codes for this analysis.

6. The yCRC_FD trained model's poorer performance in oCRC_FD than in oCRC_GZ/yCRC_GZ raises concerns about potential overfitting problems. Further investigation and clarification on this matter are necessary.

Response: We appreciate the reviewer's keen observation. We have conducted a careful re-evaluation of our machine learning experiments. In the revised manuscript, we applied a stricter criterion to filter out features only presented in only a single group. Upon closer examination, we identified a slightly inflated AUROC when the model trained on oFD was used to predict yFD (**Figure 4**, ten-fold cross-validation AUROC=0.7671, yFD prediction AUROC=0.7688). In our ten-fold cross-validation, the model was trained on 90% of the training samples and applied to the hold-out 10% samples. In the prediction step, the model was trained on 100% training samples once and applied to the testing samples. This distinction could explain why the cross-validation AUROC was lower than the AUROC observed in the testing samples.

7. There appears to be a sudden transition from displaying prediction models to pathway profiles. To improve the flow of the manuscript, the authors should ensure a smoother transition between these topics.

Response: We appreciate the reviewer's feedback, and in response, we have reorganized the manuscript. Specifically, we moved the functional analysis section before the machine learning section (lines 254-310). And we relocated the pathway-based prediction models to the machine learning section (lines 339-343). We believe these revisions lead to a better flow and transition between the different topics in the manuscript.

8. In the discussion of microbiome-based classification, the authors claimed that "Our study highlights the potential advantages of microbiome-based CRC prediction for screening." However, most of the models in this manuscript did not outperform the previous studies cited in the methods (Thomas et al., 2019; Beghini et al., 2021). This claim may be too strong based on the results and should be appropriately revised to accurately reflect the findings.

Response: We have updated the discussion at lines 433-435. The sentences read as "Microbiome-based models have been explored and strongly support the promise of non-invasive CRC diagnostics (ref 13&14). Our study expanded this promise by demonstrating that the model could be applied across a broad age range.". We have also removed the similar claim from abstract.

REVIEWER COMMENTS

Reviewer #1 (Remarks to the Author):

The report has been significantly enhanced by the authors, who have improved the analysis with a more focused approach. Specifically, they delved into strain-level identification and gene-marker based analysis in both young and old-onset colorectal cancer (CRC) settings. The results indicate similar microbial patterns at taxonomic and selected gene marker levels in both settings, corroborating previous reports that established strong associations between bacterial taxonomies, specific bacterial genes, and gene products with the onset and progression of colorectal cancer.

However, despite these improvements, the report lacks discussion on the biological and clinical implications of these findings. While I recognize that this was not the primary focus, it seems to be a missed opportunity to omit this critical context. Given the substantial number of samples sequenced and the inclusion of multiple cohorts, including controls, it would be intriguing, and indeed groundbreaking, to explore the diversity of microbial biomarkers (known and novel) and their host species, and how such diversity influences colorectal cancer outcomes.

I suggest supplementing the existing analysis (weighted) with a presence/absence-based analysis (unweighted) to identify specific markers associated with young and old-onset colorectal cancer, using both taxonomies and marker-genes. Additionally, I recommend incorporating dedicated differential-abundance algorithms such as ANCOMBC2, MaAsLin2, etc. These algorithms can be instrumental in investigating the differences in microbiome features (including taxonomies and selected marker genes) between the groups of interest. This approach would provide a more comprehensive understanding of the microbial dynamics associated with colorectal cancer across different age groups.

The writing could be significantly improved. There are lingering typos in the reported names of bacterial taxonomies. Additionally, enhancing figures by increasing symbol size and using colors with greater contrast would improve clarity.

Reviewer #2 (Remarks to the Author):

The authors carefully addresses my comments. The manuscript significantly improved. I have no major comments, but have only one minor comment.

The authors mentioned that clinical and tumor characteristics of the the Fudan cohort were previously published in Supplementary Table 2 in Yang et al. This is not good for readers. It would be great if the authors could show the clinical and tumor characteristics of the Fudan cohort as a supplementary table in this manuscript in a way that avoids plagiarism with Yang et al.

Reviewer #3 (Remarks to the Author):

The author carefully responded to all reviewers' comments. The following are some minor modification suggestions:

The authors state: "account for the effect of confounding factors, we fitted a linear regression model with sex, BMI, tumor stage and location, family history of CRC and smoking as covariates. We then calculated the Spearman correlations between the residuals and age. Similar to the Shannon index, we did not observe a significant association ($P>0.1$) between richness (the number of species) and patient age, with and without the adjustment for confounders. We have updated the text accordingly on lines 112-113."

On this issue, it is suggested that when dealing with confounding factors, the authors should consider other methods of studying confounding factors, such as Mendelian randomization (MR), and conduct synthesis to give more credible results.

REVIEWER COMMENTS

Reviewer #1 (Remarks to the Author):

The report has been significantly enhanced by the authors, who have improved the analysis with a more focused approach. Specifically, they delved into strain-level identification and gene-marker based analysis in both young and old-onset colorectal cancer (CRC) settings. The results indicate similar microbial patterns at taxonomic and selected gene marker levels in both settings, corroborating previous reports that established strong associations between bacterial taxonomies, specific bacterial genes, and gene products with the onset and progression of colorectal cancer.

Response: We thank the reviewer for excellent comments that brought the best out of our paper.

However, despite these improvements, the report lacks discussion on the biological and clinical implications of these findings. While I recognize that this was not the primary focus, it seems to be a missed opportunity to omit this critical context. Given the substantial number of samples sequenced and the inclusion of multiple cohorts, including controls, it would be intriguing, and indeed groundbreaking, to explore the diversity of microbial biomarkers (known and novel) and their host species, and how such diversity influences colorectal cancer outcomes.

I suggest supplementing the existing analysis (weighted) with a presence/absence-based analysis (unweighted) to identify specific markers associated with young and old-onset colorectal cancer, using both taxonomies and marker-genes. Additionally, I recommend incorporating dedicated differential-abundance algorithms such as ANCOMBC2, MaAsLin2, etc. These algorithms can be instrumental in investigating the differences in microbiome features (including taxonomies and selected marker genes) between the groups of interest. This approach would provide a more comprehensive understanding of the microbial dynamics associated with colorectal cancer across different age groups.

Response: We appreciate the reviewer's insightful comment on the potential biological and clinical implications of our findings. In response to the suggestion, we have expanded our analysis using ANCOMBC2 and MaAsLin2 to identify associations between microbial features and CRC characteristics.

We have added unweighted analysis based on the presence/absence of taxa and pathways to identify age-associated microbial features. The unweighted analysis, while confirming some findings from the weighted analysis, did not yield new insights.

We investigated the associations between 18 taxonomic markers (those highlighted in **Figure 2, S1, S2 & S3**) and CRC characteristics. Rigorous testing using MaAsLin2 and ANCOMBC2 was conducted for these markers. Notably, we found associations between the abundances of *Fn* and *Fa* with mismatch repair (MMR) and HER2 mutation statuses. This novel finding has been incorporated into a new **Figure 4**.

In addition, we tested the associations of gene-markers (*fadA*, *bft*, *pks* and *bai*) with tumor stage, location, and molecular characteristics. However, no significant associations were observed.

These additional analyses and findings are presented in a new section titled “Associations between microbial markers and CRC characteristics” (lines 311-333). Discussion (lines 420-427) and Methods (lines 494-497, 566-571, 705-715) have been updated accordingly.

The writing could be significantly improved. There are lingering typos in the reported names of bacterial taxonomies. Additionally, enhancing figures by increasing symbol size and using colors with greater contrast would improve clarity.

Response: We have carefully reviewed and corrected the typos in the reported names of bacterial taxonomies (*Peptostreptococcus stomatis*, *Eubacterium rectale* and *Erysipelatoclostridium ramosum*). Additionally, we have enhanced the figures by increasing symbol size and using colors with greater contrast. Specifically, we referred to the font and symbol size specifications outlined in <https://github.com/biobakery/Maaslin2/blob/master/R/viz.R>, aligning with the Nature journal’s guidelines.

Reviewer #2 (Remarks to the Author):

The authors carefully addresses my comments. The manuscript significantly improved. I have no major comments, but have only one minor comment.

Response: We sincerely thank the reviewer for very insightful and deep comments that led to a substantially improved presentation of the paper.

The authors mentioned that clinical and tumor characteristics of the Fudan cohort were previously published in Supplementary Table 2 in Yang et al. This is not good for readers. It would be great if the authors could show the clinical and tumor characteristics of the Fudan cohort as a supplementary table in this manuscript in a way that avoids plagiarism with Yang et al.

Response: We have included the phenotypic information of the Fudan cohort in our supplementary table, with clear acknowledgment that this information was derived from Yang et al. The table has been relocated and is now presented as **Table S2**.

Reviewer #3 (Remarks to the Author):

The author carefully responded to all reviewers' comments. The following are some minor modification suggestions:

Response: We thank the reviewers for their careful reading of our manuscript and making several important comments that significantly enhanced the clarity and quality of our manuscript.

The authors state: "account for the effect of confounding factors, we fitted a linear regression model with sex, BMI, tumor stage and location, family history of CRC and smoking as covariates. We then calculated the Spearman correlations between the residuals and age. Similar to the Shannon index, we did not observe a significant association ($P>0.1$) between richness (the number of species) and patient age, with and without the adjustment for confounders. We have updated the text accordingly on lines 112-113."

On this issue, it is suggested that when dealing with confounding factors, the authors should consider other methods of studying confounding factors, such as Mendelian randomization (MR), and conduct synthesis to give more credible results.

Response: We appreciate the suggestion to explore methods like MR for studying confounding factors. However, current genomic studies have not identified robust genetic instruments for CRC-associated markers, limiting the application of MR in our analysis. This limitation is acknowledged in Discussion (lines 453-456).

REVIEWERS' COMMENTS

Reviewer #1 (Remarks to the Author):

In this round of revisions, the authors have addressed most of the comments and concerns raised but the reviewers during the previous rounds of revisions. Although the suggested analysis looking into microbial hallmarks of colorectal cancer with clinical outcomes was not presented, the publication of the current results and associated data will add an important contribution to the field.